# *4D-VLA*: Spatiotemporal Vision-Language-Action Pretraining with Cross-Scene Calibration

Jiahui Zhang[1][*]    Yurui Chen[1][*]    Yueming Xu[1]    Ze Huang[1]    Yanpeng Zhou[2]
Yu-Jie Yuan[2]    Xinyue Cai[2]    Guowei Huang[2]    Xingyue Quan[2]    Hang Xu[2]    Li Zhang[1][†]

[1] School of Data Science, Fudan University    [2] Huawei Noah's Ark Lab

https://github.com/LogosRoboticsGroup/4D-VLA

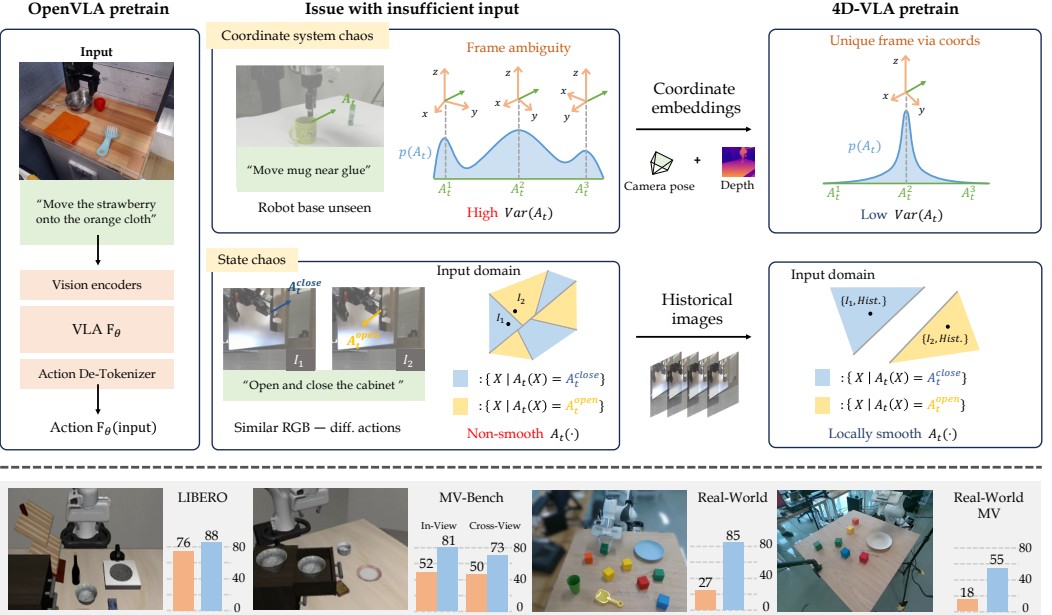

Figure 1: **Top:** Our pretraining design philosophy highlights that prior methods often lack key cues in their input for accurate action inference. This leads to target action distributions $A_t(\cdot)$ exhibiting high variance or non-smoothness, which negatively impacts pretraining performance. A rough analysis shows that in the DROID dataset, 67% of the samples have the robot's base occluded, causing coordinate system chaos. **Bottom:** We verify our method in both simulated and real-world robotic settings and report the performance for the OpenVLA baseline and our *4D-VLA* approach.

## Abstract

Leveraging diverse robotic data for pretraining remains a critical challenge. Existing methods typically model the dataset's action distribution using simple observations as inputs. However, these inputs are often incomplete, resulting in a dispersed conditional action distribution—an issue we refer to as coordinate system chaos and state chaos. This inconsistency significantly hampers pretraining efficiency. To address this, we propose *4D-VLA*, a novel approach that effectively integrates 4D information into the input to mitigate these sources of chaos. Our model introduces

---

[*]Equal contribution. [†]Corresponding author (lizhangfd@fudan.edu.cn).

39th Conference on Neural Information Processing Systems (NeurIPS 2025).

depth and temporal information into visual features with sequential RGB-D inputs, aligning the coordinate systems of the robot and the scene. This alignment endows the model with strong spatiotemporal reasoning capabilities while minimizing training overhead. Additionally, we introduce *memory bank sampling* , a frame sampling strategy designed to extract informative frames from historical images, further improving effectiveness and efficiency. Experimental results demonstrate that our pretraining method and architectural components substantially enhance model performance. In both simulated and real-world experiments, our model achieves a significant increase in success rate over OpenVLA [1]. To further assess spatial perception and generalization to novel views, we introduce *MV-Bench*, a multi-view simulation benchmark. Our model consistently outperforms existing methods, demonstrating stronger spatial understanding and adaptability.

# 1    Introduction

The emergence of pretrained vision-language models has established an effective framework for aligning human language with visual data, advancing embodied intelligence. Meanwhile, the open-sourcing of diverse robotic datasets [2, 3, 4, 5] has enabled data-driven robot control. However, efficiently extracting useful information from these datasets remains a challenge for improving generalization across diverse scenarios.

A real-world action distribution can be interpreted as a response function conditioned on observations or input, denoted as $A_t(\text{input})$. The objective of pretraining is to learn a model $F_\theta(\text{input})$ that approximates this function using large-scale data $A_t^{data}(\text{input})$. However, existing pretraining paradigms often suffer from incomplete or under-informative input, lacking critical contextual cues required for reliable action reasoning. Consequently, the resulting target distribution $A_t(\cdot)$ may exhibit undesirable characteristics—such as lack of smoothness, high variance, or multimodality—which hinder the model's ability to learn robust and generalizable behaviors.

Previous approaches, such as OpenVLA [1], use only a single RGB image and a textual instruction as input. This limited input setting leads to two prominent types of chaos. The first is coordinate system chaos, which arises when actions are defined in the robot's coordinate frame, yet the visual input lacks sufficient spatial context. For instance, if the image does not fully capture the robot's body, it becomes challenging to infer the robot's exact position and orientation. The second is state chaos, which arises in scenarios where a single frame lacks the necessary temporal or contextual cues to resolve action ambiguity. This includes symmetric trajectories—where it is difficult to infer the direction of motion—as well as cases where visually similar observations correspond to entirely different actions. These ambiguities hinder the effectiveness of pretraining, as illustrated in Fig. 1. In contrast, HPT [6] extends the input by incorporating some dataset-specific parameters, which helps address the issue of inconsistent coordinate systems across different datasets. However, this approach lacks scalability and increases the complexity of training.

To address this, we propose *4D-VLA*, a framework that integrates 4D spatiotemporal information to resolve such ambiguities. By combining spatial coordinate embeddings with a 3D-aware module, it generates spatial vision tokens that align the robot's coordinate system with the scene, enhancing 3D perception. This also enables efficient encoding of multiple historical frames, improving temporal reasoning. Additionally, we introduce *memory bank sampling*, a frame sampling strategy that selects key frames based on historical similarity, boosting model efficiency.

To further investigate the spatial understanding and generalization of VLA models, we introduce *MV-Bench*, a multi-view dataset that evaluates performance across diverse viewpoints. Our approach enables robust pretraining, improving generalization to novel scenarios while outperforming baselines.

Our contributions are: **(i)** We propose *4D-VLA*, an efficient VLA model that integrates a spatial module with vision features to generate 3D-aware spatial vision tokens, effectively mitigating coordinate system and state chaos, thereby significantly enhancing pretraining efficiency. Additionally, we introduce *memory bank sampling*, a simple but effective method for historical information sampling. **(ii)** Our model has been validated both in the simulated and real-world environment, demonstrating its superiority. **(iii)** We develop a multi-view simulation dataset *MV-Bench* to evaluate spatial understanding and generalization, on which our model achieves outstanding performance.

## 2   Related works

**Vision-language models** Recent advancements in vision-language models (VLMs) have significantly enhanced the integration of vision and language understanding across diverse domains. Models like Flamingo [7], LLAVA [8], and BLIP-2 [9] focus on aligning text and image feature spaces to facilitate effective image understanding. More recently, there has been growing interest in expanding language models to support multi-image inputs, enabling them to handle more complex tasks and real-world scenarios. For instance, [10, 11, 12, 13] utilize multi-image sequences to capture temporal and action-related dynamics across frames, providing robust video comprehension. In addition, some recent models have begun incorporating 3D information to bolster spatial reasoning, as seen in 3D-LLM [14] and Scene-LLM [15]. These models leverage 3D inputs to enhance understanding of spatial relationships within a scene. However, none of these models explicitly integrate 4D spatiotemporal information to fully capture both spatial and temporal dynamics within the architecture.

**Vision-language-action models** The vision-language-action (VLA) model represents a significant advancement in vision-language research, enabling more complex and interactive tasks aimed at facilitating real-world environment interactions. [16, 17, 18] complete tasks by directly predicting trajectories. [19, 20, 21, 22, 23, 24] predict the robot's current actions to enable closed-loop control, while [25, 26] enhance action prediction by training a world model to forecast future states. Some works [27, 28] also drives VLA policies using simple concatenation of historical observations.

Recent works leverage diverse robotic datasets from various scenes and robot types to pretrain models for better generalization in novel environments. [29, 30, 3, 1] use the single image as input and are pretrained on large-scale datasets, while Octo [31] incorporates historical context and uses a diffusion head to predict the next $n$ actions. HPT [6] addresses the heterogeneity among different datasets by introducing dataset-specific parameters during pretraining to improve training efficiency. However, these methods overlook that the inefficiency in prior pretraining arises from insufficient input context, resulting in a high variance of the conditioned action distribution $A_t(\cdot)$ and ultimately hindering pretraining effectiveness. Our approach tackles this issue by introducing 4D information to mitigate coordinate system chaos and state chaos. This enables the model to learn meaningful action distributions from diverse datasets, thereby enhancing performance.

## 3   Method

This section provides a comprehensive overview of our proposed *4D-VLA*. As shown in Fig. 2, our model processes sequential RGB-D images as input, converting them into corresponding spatial vision tokens. These tokens, together with task-specific text tokens, serve as feature inputs for the subsequent Transformer. After decoding through the VLM Transformer and action head, it ultimately generates the action output.

### 3.1   Preliminary

**Problem definition** The vision-language action (VLA) model takes a language instruction as input and aims to control a robot to accomplish the specified task. Specifically, VLA with a low-level control policy refers to a class of models that use the current observations as input to predict an action for the robot in its present state, enabling end-to-end, closed-loop control. The action is defined by three components: $\Delta\boldsymbol{x} \in \mathbb{R}^3$, $\Delta\boldsymbol{\theta} \in \mathbb{R}^3$, and $g \in [0, 1]$, representing the control translation, rotation offset, and the open-close state of the robot's end-effector, respectively.

**Vision-language model backbone** We leverage a pretrained large vision-language model (VLM) as the backbone, specifically InternVL-4B [12], which consists of a text tokenizer $\mathcal{T}$, a vision encoder $\mathcal{E}$, and a Transformer decoder $\mathcal{D}$. The vision encoder processes visual observations, which are subsequently compressed by an MLP projector $\mathcal{P}$ to generate vision embeddings, while text inputs are tokenized and embedded to form structured textual tokens. These multimodal tokens are then fed into the decoder $\mathcal{D}$ for next-token prediction. This backbone provides a robust foundation for aligning visual information with the shared semantic space, enabling effective robot action generation.

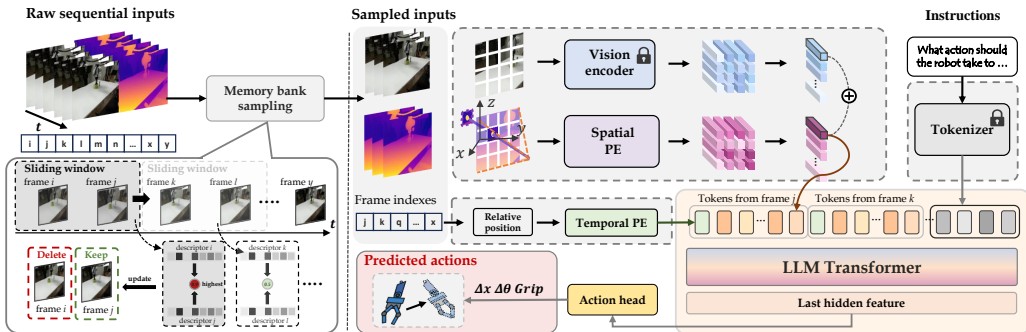

Figure 2: **Our *4D-VLA* pipeline.** Our *memory bank sampling* method selects informative frames from sequential RGB-D inputs. A vision encoder with 3D coordinate embeddings generates spatial-aware tokens, which are fused into a 4D spatiotemporal representation. Combined with text tokens, these are processed by the LLM to decode actions via an action head.

## 3.2 Spatial-aware visual tokens

A reasonable action prediction requires awareness of both semantic perception and spatial perception of the scene. A fundamental aspect of spatial perception is that the robot must accurately determine its relative position within the scene, effectively resolving coordinate system chaos. To address this issue while enhancing spatial perception without compromising semantic awareness, we fuse the coordinate information derived from depth with vision tokens, resulting in spatial vision tokens.

In our method, the input image $\mathbf{I} \in \mathbb{R}^{3 \times h \times w}$ is first encoded by $\mathcal{E}$ into a feature map with a downsampling rate of $c$, yielding $\mathbf{f}_v = \mathcal{E}(\mathbf{I}) \in \mathbb{R}^{k \times \frac{h}{c} \times \frac{w}{c}}$, where $k$ denotes the feature dimension. Next, we obtain a downsampled depth map $\mathbf{D} \in \mathbb{R}^{\frac{h}{c} \times \frac{w}{c}}$, which assigns depth values to each corresponding feature volume. Using the camera's extrinsic $[\mathbf{R}|\mathbf{T}]$ and intrinsics $\mathbf{K}$, we back-project the depth value into 3D coordinates $\mathbf{P}_w \in \mathbb{R}^{3 \times \frac{h}{c} \times \frac{w}{c}}$ within the world (or robot) coordinate system:

$$\mathbf{P}_w(\cdot, u, v) = \mathbf{R} \left( \mathbf{D}(u, v) \cdot \mathbf{K}^{-1} \begin{bmatrix} u \\ v \\ 1 \end{bmatrix} \right) + \mathbf{T}. \tag{1}$$

We apply a learnable positional embedding $\mathcal{E}_S$ to encode the 3D coordinates and integrate it with the original visual feature map via element-wise addition, forming spatial vision features that enhance spatial representation. These features are then processed by the MLP projector $\mathcal{P}$ within InternVL [12], generating spatial vision tokens $e^{ST} = \mathcal{P}(\mathcal{E}(\mathbf{I}) + \mathcal{E}_S(\mathbf{P}_w))$.

## 3.3 4D representation with multi-frame encoding

As spatial vision tokens are designed to represent information from a single frame, and the LLM is fine-tuned with video data, we can naturally extend our encoding process to incorporate multi-frame information using the same approach. This allows us to integrate coherent 4D spatiotemporal representations seamlessly. A naive approach is to use the spatial vision tokens from the current frame along with those from uniformly sampled historical frames as input $\{e_{t-j}^{ST} \mid j = 0, 1, 2, \ldots, n-1\}$, where $n$ denotes the total number of temporal frames used, and $t$ represents the index of the current frame in the sequence. These tokens, together with the corresponding text tokens from task instructions, are then fed into the VLM's Transformer for decoding.

**Memory bank sampling.** However, our experiments reveal that the performance of the model is highly sensitive to both the sampling interval and the total temporal window $n$. While denser frame sampling improves performance, it significantly increases memory consumption and reduces inference speed. Moreover, since a robot's movement speed varies over time, a naive uniform sampling strategy often leads to redundant information, resulting in inefficiencies.

To effectively exploit temporal information, we propose an adaptive historical frame sampling method based on a memory bank, aiming to capture rich historical information with a minimal number of frames. Specifically, given current timestamp $t$, all image observations $\{\mathbf{I}_{t-j} \mid j = 0, 1, 2, \ldots, n-1\}$ with a temporal window $n$, *memory bank sampling* $\mathcal{M}$ returns a set of $k$ sampled timestamps $\mathcal{H} = \mathcal{M}(t, \{\mathbf{I}\}, k, \phi)$, where $k$ is the number of sample frames based on certain feature extractor $\phi$.

The algorithm is detailed in Alg. 1, which sequentially traverses image groups while maintaining a similarity queue, ensuring each newly added frame has lower similarity than the current maximum.

**Temporal positional encoding.** Since *memory bank sampling* follows a non-uniform strategy, it is essential to encode the temporal position of each sampled spatial vision token relative to the current frame. To enhance the model's flexibility and generalization, we introduce a time encoding token $e^T$, which captures the relative temporal offset between the historical and current frames, $e_j^T = \mathcal{E}_T(t - j)$, where $j$ denotes the timestamp of a historical frame, $t$ represents the current observation time, and $\mathcal{E}_t$ is the learnable temporal encoding function. Accordingly, the final input token set is structured as:

$$\mathcal{X} = \bigcup_{i \in \mathcal{H}} [e_i^T \mid e_i^{ST}] \cup \{e^{text}\}, \tag{2}$$

where $e^{text}$ represents the text instruction tokens. Each sampled frame $i$ contributes a token pair $[e_i^T \mid e_i^{ST}]$, with the temporal encoding token preceding the spatial vision token. This design ensures that the model effectively captures temporal relationships while maintaining spatial-awareness.

## 3.4 Loss functions

To accelerate control policy generation, we use an action head with two MLP layers that predict the action $[\Delta\hat{x}, \Delta\hat{\theta}, \hat{g}]$ using the hidden features of the VLM Transformer's last token.

Our total training loss can be written as follows:

$$\mathcal{L} = \mathcal{L}_t + \mathcal{L}_r + \mathcal{L}_g + \lambda_d \mathcal{L}_d, \tag{3}$$

where the translation loss $\mathcal{L}_t = \|\Delta\hat{x} - \Delta x\|_2$, the rotation loss $\mathcal{L}_r = \|\Delta\hat{\theta} - \Delta\theta\|_2$, and the grip loss $\mathcal{L}_g = \text{BCE}(\hat{g}, g)$. Since the translation $\|\Delta x\|$ in action is often small, we place greater emphasis on directional awareness within the action by introducing a directional loss:

$$\mathcal{L}_d = \|d(\Delta\hat{x}) - d(\Delta x)\|_2, \tag{4}$$

where $d(x) = \frac{x}{\|x\|_2 + \epsilon}$. $\epsilon$ is a small value ensuring smoothness at zero.

---

**Algorithm 1:** *memory bank sampling*

**Input:** $t, \{\mathbf{I}_{t-j} \mid j = 0, 1, \ldots, n-1\}$, sample size $k$,
       feature extractor $\phi$
**Output:** A set of sampled timestamps $\mathcal{H}$
Initialize $\mathcal{H} \leftarrow [t]$     ▷ ***Start with the current frame***
Initialize $\mathbf{S} \leftarrow [-\inf]$     ▷ ***Similarity list***
**for** $j = 1$ to $n - 1$ **do**
    $s = \text{Similarity}(\phi(\mathbf{I}_{\mathcal{H}[-1]}), \phi(\mathbf{I}_{t-j}))$
    **if** $\text{len}(\mathcal{H}) < k$ **then**
        Append $t - j$ to $\mathcal{H}$; Append $s$ to $\mathbf{S}$
    **else**
        $m = \arg\max(\mathbf{S})$
        **if** $s < \mathbf{S}[m]$ **then**   ▷ ***Insert and reorganize***
            Append $t - j$ to $\mathcal{H}$; Append $s$ to $\mathbf{S}$
            $s' = \text{Similarity}(\phi(\mathbf{I}_{\mathcal{H}[m-1]}), \phi(\mathbf{I}_{\mathcal{H}[m+1]}))$
            Remove $\mathcal{H}[m]$
            Replace $\mathbf{S}[m+1]$ with $s'$; Remove $\mathbf{S}[m]$
        **else**         ▷ ***Replace the last frame***
            $s' = \text{Similarity}(\phi(\mathbf{I}_{\mathcal{H}[-2]}), \phi(\mathbf{I}_{\mathcal{H}[t-j]}))$
            Remove $\mathcal{H}[-1]$
            Append $t - j$ to $\mathcal{H}$; Append $s'$ to $\mathbf{S}$
**return** $\mathcal{H}$

---

## 3.5 *MV-Bench*

We propose the *MV-Bench* to provide a comprehensive evaluation of model capabilities in learning control policies across diverse viewpoints and generalizing to novel views, while also assessing the model's spatial understanding and adaptability.

**Benchmark settings.** We build a multi-view dataset based on LIBERO-SPATIAL [4]. For each trajectory, we sample 6 training and 6 testing viewpoints uniformly within a 270° front-facing range. Evaluation includes two tasks: *In-View*, where training and testing use the same views; and *Cross-View*, where testing is done on unseen viewpoints. Our camera settings are shown in Fig. 3.

## 4 Experiments

We first introduce the datasets and simulation environment, then describe pretraining and fine-tuning. Our model is pretrained on real-world data and fine-tuned with both simulation and real-world trajectories. We run closed-loop evaluations in diverse environments and report task performance.

In addition, we evaluate on the ARM4R [32] benchmark for a direct comparison with ARM4R. Note that ARM4R pretrains with 3D inputs, while our approach differs in supervision and architecture; results and a detailed discussion are provided in Appx. 7.1.

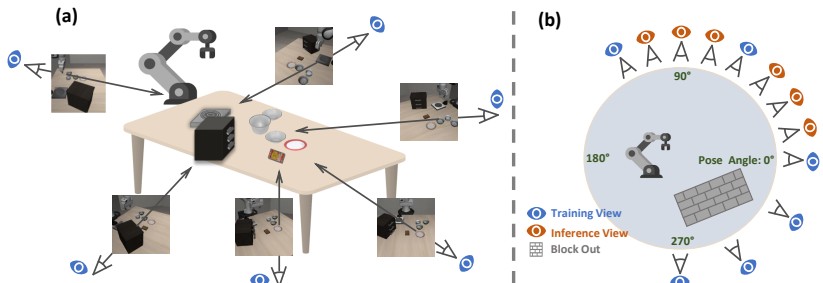

Figure 3: **Our *MV-Bench* camera setting.** We select 6 diverse viewpoints as training views and render images for all LIBERO-SPATIAL tasks. Novel inference views are placed near the training views. To avoid occlusion from the black box, test views in blocked areas are excluded.

| Method | Spatial | Object | Goal | Long | Avg. |
|---|---|---|---|---|---|
| UniAct-0.5B [33]† | 64.5 | 77.5 | 68.0 | 46.5 | 64.1 |
| SparseVLM [34]† | 79.8 | 67.0 | 72.6 | 39.4 | 64.7 |
| FastV [35]† | 83.4 | 84.0 | 74.2 | 51.6 | 73.3 |
| VLA-Cache [36]† | 83.8 | 85.8 | 76.4 | 52.8 | 74.7 |
| DiffusionPolicy [16] | 78.3 ± 1.1 | 92.5 ± 0.7 | 68.3 ± 1.2 | 50.5 ± 1.3 | 72.4 ± 0.7 |
| TraceVLA [37] | 84.6 ± 0.2 | 85.2 ± 0.4 | 75.1 ± 0.3 | 54.1 ± 1.0 | 74.8 ± 0.5 |
| SpatialVLA [38] | 88.2 ± 0.5 | 89.9 ± 0.7 | 78.6 ± 0.6 | 55.5 ± 1.0 | 78.1 ± 0.7 |
| Octo [31] | 78.9 ± 1.0 | 85.7 ± 0.9 | 84.6 ± 0.9 | 51.1 ± 1.3 | 75.1 ± 0.6 |
| OpenVLA [1] | 84.7 ± 0.9 | 88.4 ± 0.8 | 79.2 ± 1.0 | 53.7 ± 1.3 | 76.5 ± 0.6 |
| ***4D-VLA*(Ours)** | **88.9** ± 0.5 | **95.2** ± 0.3 | **90.9** ± 0.4 | **79.1** ± 1.2 | **88.6** ± 0.3 |

Table 1: **Evaluation of success rate on LIBERO.** Bold indicates the best-performing model. Our model significantly outperforms other competitors, with an average success rate 12.1 higher than OpenVLA. †Denotes no available standard deviation data.

## 4.1 Datasets and simulation environments

**DROID [2]** A diverse real-world robot manipulation dataset with 76,000 demonstration trajectories, or 350 hours of interaction data, spanning a total of 564 scenes and 86 tasks, each featuring RGB-D data from two third-person and one wrist-mounted camera.

**LIBERO [4]** The LIBERO benchmark is a simulation suite with 4 task sets designed to advance life-long learning in robotic manipulation. LIBERO-SPATIAL, LIBERO-OBJECT, and LIBERO-GOAL explore knowledge transfer in spatial reasoning, object understanding, and task goals. LIBERO-100 includes 90 short-horizon (LIBERO-90) and 10 long-horizon (LIBERO-LONG) tasks, covering 130 subtasks, each with 50 trajectories captured from both a main and wrist-mounted camera.

## 4.2 Pretraining setups

**Data process.** We pretrain our model on the DROID [2] dataset. RGB-D frames are resized to $448 \times 252$, and each trajectory is uniformly downsampled to 100 actions. We remove frames with unchanged proprioception, specifically the stationary frames, and exclude trajectories with a total action count exceeding 600. Actions are defined as the difference between the end-effector's current and target states, with translation scaled by 15 and rotation (Euler angles) by 5 for normalization.

**Pretraining details.** Our pretrained model is based on InternVL-4B. We use a temporal window of 20 and apply *memory bank sampling* to select 5 past frames along with the current frame. The RGB-D inputs are processed by the original vision encoder in InternVL, with 3D positional encodings incorporated before being passed to a projector—an MLP with a downsampling rate of 4, initialized from the pretrained InternVL weights. To handle sparse or incomplete depth from DROID, we compute the mean depth within each vision patch. If over 90% of a patch is masked, we skip adding 3D features, effectively acting as dropout and data augmentation.

In the training process, we freeze the vision encoder but training all other parameters. $\lambda_d$ is set to 1. We utilize a cosine learning rate scheduler with an initial learning rate of 2e-5. Our model was trained for 1 epoch with a batch size of 512, requiring around 20k steps to complete. Training was conducted on 8 NVIDIA A6000 GPUs over 96 hours. Inference with FlashAttention in bf16 requires approximately 8 GB of GPU memory.

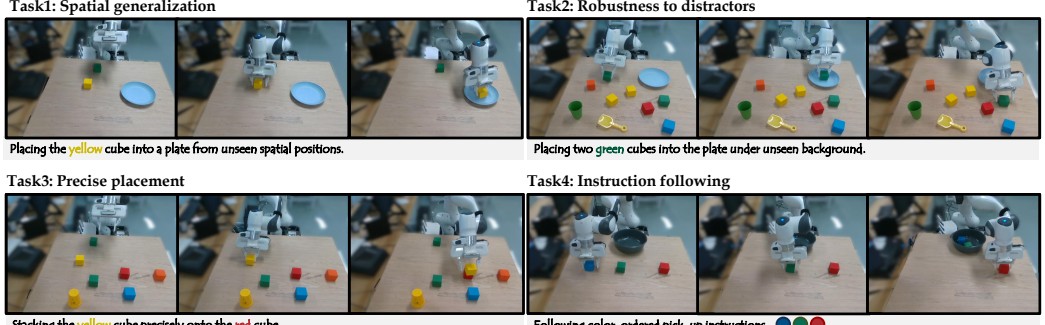

Figure 4: **Our real-world experiment settings.** These settings aim to evaluate the model's spatial generalization, robustness to distractors, precision in placement, and ability to follow instructions. Each row presents a 3-frame execution snapshot.

## 4.3 LIBERO evaluation

After pretraining, we fine-tune and conduct close-loop testing in the LIBERO simulation environment, using task success rate as the metric to evaluate the performance.

**Evaluation protocol.** For each task in LIBERO, there are 10 test subtasks, and each subtask contains 50 different object layouts. This requires 1500 simulation tracks to evaluate a single task. The random seed is used to alter the initial state of the objects. To evaluate each task, we randomly sample 3 different seeds to calculate the mean and standard deviation of the success rate.

**Fine-tuning details.** Unlike the pretraining phase, we used the simplest input settings to enable our model to learn the interaction effects between 3D information and historical data on actions. In the subsequent fine-tuning phase, we set the number of sequential frames $k = 5$ with a window size $n = 20$. The current state RGB-D image is resized to 448 $\times$448, while historical images are resized to 224$\times$224 to optimize memory usage and model efficiency. Following [1], we normalize the ground truth action label using the 99th and 1st percentiles. We employed a cosine learning rate scheduler with a learning rate of 4e-5, using a batch size of 128 and training for 20 epochs. $\lambda_d$ is set to 0. During training, we activated all network parameters except the vision encoder.

**Experimental results.** As shown in Tab. 1, our *4D-VLA* shows clear performance gains over prior methods, with 5.2% over OpenVLA[1] on LIBERO-SPATIAL, 2.7% over DiffusionPolicy [16] on LIBERO-OBJECT, and 6.3% over Octo [31] on LIBERO-GOAL. On the most challenging LIBERO-LONG task, it surpasses OpenVLA by 25.4%. On average, *4D-VLA* improves success rate by 12.1% than OpenVLA, demonstrating stronger stability and spatiotemporal reasoning in complex settings.

## 4.4 *MV-Bench* evaluation

As shown in Tab. 2, our model achieves a 81.0% success rate in the *In-View* setting, demonstrating its capability to handle diverse training views effectively. This result shows that the integration of spatiotemporal information enables our model to manage challenging and conflicting perspectives, outperforming OpenVLA [1] by a significant margin of 28.8%. In the *Cross-View* evaluation, our model also achieves the best performance, indicating a notable generalization capability across novel viewpoints. It highlights the robustness of our model in handling diverse viewpoints.

| Method | In-View, $\Delta 0°$ | | | | | | Avg. |
|---|---|---|---|---|---|---|---|
| *Angle* | 0° | 60° | 120° | 270° | 300° | 330° | |
| OpenVLA [1] | 57.4 | 50.0 | 50.6 | 43.5 | 53.8 | 57.8 | 52.2 |
| *4D-VLA* (**Ours**) | 83.2 | 87.0 | 79.5 | 70.2 | 75.8 | 90.2 | **81.0** |
| | Cross-View, $\Delta 15°$ | | | | $\Delta 30°$ | | |
| *Angle* | 15° | 45° | 75° | 105° | 30° | 90° | |
| OpenVLA [1] | 64.0 | 48.2 | 54.2 | 34.2 | 63.0 | 39.2 | 50.5 |
| *4D-VLA* (**Ours**) | 83.4 | 83.2 | 74.0 | 65.6 | 75.8 | 60.8 | **73.8** |

Table 2: **Evaluation of success rate on *MV-Bench*.** $\Delta$ symbol representing the angular deviation from the nearest training viewpoint along the z-axis.

## 4.5 Real-world evaluation

To evaluate models in real-world scenarios, we conducted physical experiments using a Franka robotic arm. Specifically, we designed 4 representative tasks to assess the model's capabilities, using success rate as the primary evaluation metric. For each task, we manually collected 50 diverse trajectories for

| Method | Task 1 | Task 2 | Task 3 | Task 4 | Avg. |
|---|---|---|---|---|---|
| OpenVLA [1] | 45.00 | 22.50 | 30.00 | 13.33 | 27.70 |
| Base VLA | 35.00 | 20.00 | 5.00 | 2.67 | 15.67 |
|   + Pretraining | 60.00 | 60.00 | 40.00 | 28.00 | 47.00 |
|   + Pretraining + Coord. | 75.00 | 60.00 | 85.00 | 34.67 | 63.67 |
|   + Pretraining + Hist. | 80.00 | 77.50 | 70.00 | 36.00 | 65.88 |
|   + Pretraining + Coord. + Hist. *(full model)* | **90.00** | **82.50** | **90.00** | **80.00** | **85.63** |

Table 3: **Real-world evaluation results.** We incrementally improve the Base VLA by adding pretraining, coordinate encoding, and historical frames selected via *memory bank sampling* .

training and trained the model for 20 epochs. For Task 4, involving color-ordered pick-up instructions, we used 5 color sequences with 10 demonstrations each, totaling 50 trajectories.

**Task descriptions.** As shown in Fig. 4, we design 4 real-world manipulation tasks to evaluate different aspects of the model's spatial reasoning and generalization capabilities. **Task 1: Spatial generalization.** The robot is tasked with placing a yellow cube into a plate from positions not seen during training, evaluating its ability to generalize across novel spatial positions. **Task 2: Robustness to distractors.** The robot is asked to place two green cubes into the plate in scenes with cluttered backgrounds. This task evaluates the model's robustness to environments distractors. **Task 3: Precise placement.** The robot is asked to precisely stack the yellow cube onto the red one. This task emphasizes the need for fine-grained action prediction. **Task 4: Instruction following.** The robot is asked to execute color-ordered pick-and-place commands (e.g., red → green → blue). The task assesses the model's ability to follow structured instructions correctly.

**Evaluation metrics.** Tasks 1 and 3 are evaluated by success rate (i.e., successful trials / total trials). For Task 2, each correctly placed green block earns 1 point (maximum 2 per trial), with performance measured as a total score out of 40. Task 4 involves following color-ordered instructions across 5 combinations (5 trials each). Each correctly placed block earns 1 point (maximum 3 per trial), for a total score out of 75. The final performance is reported as score divided by 75.

**Experimental results.** We set InternVL-4B with single RGB image inputs followed by an action head as our Base VLA model. Based on this, we incrementally add different modules to investigate their individual contributions to performance. OpenVLA [1] is set as our main competitor.

As shown in Tab. 3, the base VLA model without pretraining underperforms OpenVLA on all tasks. Our spatially grounded pretraining significantly boosts performance—even with a single RGB-D frame—confirming its effectiveness. In short-horizon tasks (Task 1 and 3), adding coordinate information notably improves results, especially in Task 3, which demands precise spatial alignment. This shows that coordinate encoding enhances spatial grounding and action accuracy. In long-horizon tasks (Task 2 and 4), the model often succeeds in the first step but fails the second without access to history, due to latent state ambiguity. *memory bank sampling* alleviates this by providing temporally relevant frames, improving multi-step reasoning. Overall, we find that matching the downstream input to the pretraining setting (e.g., coordinate-aware, temporally structured) leads to better transfer of spatial representations. Even with mismatched inputs, our model still outperforms baselines, showing strong generalizability. Coordinate encoding offers explicit spatial cues, while multi-frame inputs provide temporal context to disambiguate action intent.

## 4.6 Multi-view real-world evaluation

In this section, we conduct additional real-world experiments under a multi-view camera setup. We design two more challenging tasks to evaluate the model's generalization ability with respect to: **(i)** variations in object locations together with the changed background environments; **(ii)** inputs from novel camera viewpoints. We use 4 fixed cameras to capture each demonstration from different angles, collecting 50 trajectories per task per camera—resulting in a total of 200 trajectories per task for training. All models are trained for 20 epochs, and performance is measured by success rate.

**Task descriptions.** These two more challenging tasks are shown in Fig. 5. **Task 1: Out-of-distribution generalization**. The robot is tasked with placing a yellow cube into a plate under conditions where both the spatial configuration and the surrounding environment differ from those seen during training. These variations include changes in the plate's position, the presence and location

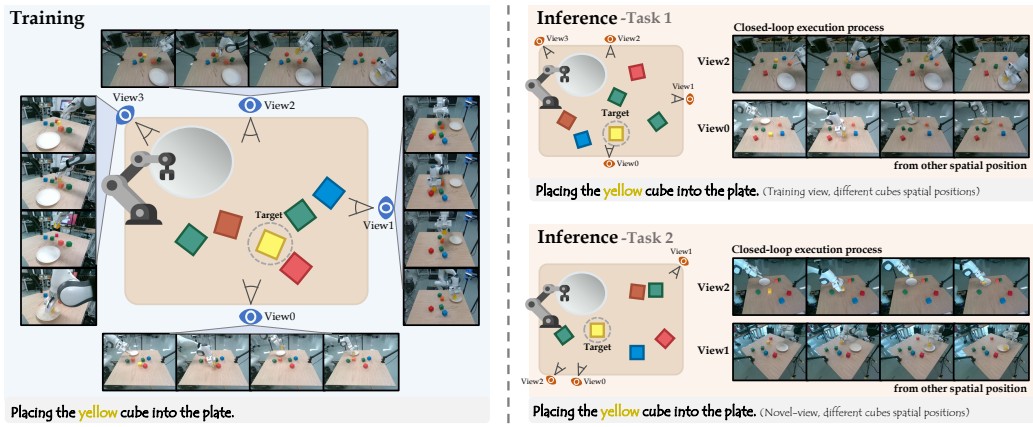

Figure 5: **Our multi-view real-world experiment settings.** These settings aim to evaluate the model's out-of-distribution and novel-view generalization ability.

| Method | In-View | | | | Cross-View | | | Avg. |
|---|---|---|---|---|---|---|---|---|
| *Angle* | 0° | 90° | 180° | 225° | $\Delta15°(-15°)$ | $\Delta25°(-25°)$ | $\Delta45°(135°)$ | |
| OpenVLA [1] | 25 | 15 | 30 | 10 | 30 | 10 | 5 | 18 |
| *4D-VLA* (Ours) | 60 | 50 | 65 | 65 | 50 | 55 | 40 | 55 |

Table 4: **Real-world multi-view evaluation.** We test our model's spatial generalization across varying viewpoints and object layouts. 4D-VLA shows strong in-view and cross-view performance, highlighting its robustness under real-world distribution shifts.

of distractor objects (e.g., other cubes). This task evaluates the model's ability to generalize to unseen object arrangements and background contexts, testing its robustness in real-world deployments beyond the training distribution. **Task 2: Novel-view generalization**. Similar to Task 1, the robot is asked to place a yellow cube into a plate, with spatial setup and surroundings differing from training. However, during inference, data input is captured exclusively from an additional novel, unseen camera viewpoint that was not used during training. This task evaluates the model's viewpoint robustness—its ability to generalize across camera perspectives and accurately interpret the scene from unfamiliar angles. Success in this task reflects strong spatial understanding and invariance to viewpoint changes, both critical for real-world multi-camera deployment. To simplify the setup, the target block is only moved within a small spatial range, while the background is fully randomized.

**Evaluation metrics.** Each multi-view task is evaluated over 20 trials. In every trial, both the background and object positions are randomly shuffled to assess the model's robustness and generalization. The evaluation metric is the task success rate, computed as the ratio of successful trials to the total number of trials.

**Experiments results.** As shown in Tab. 4, *4D-VLA* significantly outperforms OpenVLA in both in-view and cross-view settings, demonstrating strong generalization to viewpoint shifts and layout variations. In the in-view setting, where the camera is fixed but object layouts change, our model maintains consistently high success rates, indicating robustness to spatial perturbations. In the more challenging cross-view setting involving unseen viewpoints, *4D-VLA* continues to perform stably across different angles. Although performance slightly drops at larger viewpoint shifts (e.g., $\Delta45°$), it remains stable compared to OpenVLA, whose success rate fluctuates more severely under such conditions. These results suggest that our model effectively captures spatial consistency across views, leading to more reliable visuomotor control in real-world environments.

## 5  Discussion

Building upon previous experiments, we further analyze three key questions: **(i) The Role of historical information** -How does historical context influence model's effectiveness? **(ii) Ablation study on model components** -How do different architectural components contribute to the overall

performance of the model? **(iii) The impact of coordinate system chaos** -How does coordinate system inconsistency affect the model's performance, and can the introduction of a 3D representation mitigate this issue? The first question is addressed in Sec. 5.1; the second is covered in Appx. 7.2, and the third in Appx. 7.3. Our discussion begins with the simple model, i.e., InternVL-4B with an MLP action head, using single RGB image as the vision input.

### 5.1 Exploring historical information utilization

As highlighted in Octo [31], incorporating historical context can significantly boost model performance. However, this has been underexplored. We conduct a comprehensive analysis to assess its true impact on model effectiveness. We define window size $n$ as the number of historical frames available to the model and $k$ as the number of frames sampled from them. To systematically investigate their impact, we design two experimental settings: **(i)** fixing $n$ while progressively increasing $k$, and **(ii)** fixing $k$ while expanding $n$.

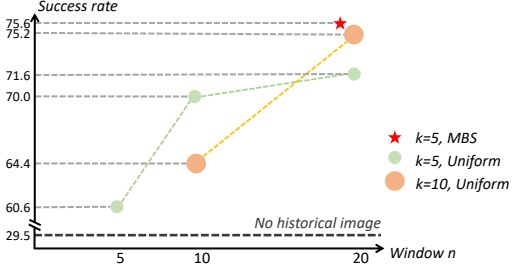

| Encoding | Position | Fusion | Success rate |
|---|---|---|---|
| learnable | relative | concat | **75.6** |
| learnable | relative | additive | 74.8 |
| sinusoidal | relative | additive | 71.6 |
| learnable | absolute | additive | 0.0 |
| *-No temporal encoding-* | | | 63.0 |

Table 5: **Ablation on temporal encoding method.**

Figure 6: **Historical image analysis.** Larger points indicate lower efficiency.

As illustrated in Fig. 6, our findings reveal that the historical window size $n$ plays a pivotal role in determining performance, whereas the effect of $k$ is comparatively minor. This suggests that uniform sampling may introduce excessive redundancy, adversely diminishing the model's efficiency. *memory bank sampling* strategy, which effectively reduces redundancy and improves performance, even with a smaller $k$, showcasing its effectiveness in maximizing the utility of historical context.

**Ablation on temporal encoding method.** Since MBS employs non-uniform sampling, the absence of explicit temporal position encodings can lead to ambiguity in the historical sequence, making it difficult for the model to fully leverage past information. To address this, we investigate the effect of different temporal encoding strategies on model performance, as summarized in Tab. 5. Specifically, *additive* means the temporal encoding is directly added to the image tokens, while *concat* appends it before the image token, and *absolute* refers to encoding based on the sampled timestamps rather than relative to the current frame. Clearly, the *concat* method achieves the best performance, with the key advantage of seamlessly integrating with spatial vision tokens to enhance the model's representation.

## 6 Conclusion

In this paper, we present *4D-VLA*, which incorporates 4D information to address challenges in leveraging diverse robotic datasets for pretraining, such as coordinate system chaos and state chaos. Our model encodes sequential RGB-D images into visual features with corresponding 3D coordinates, aligning the robot's coordinate system with the scene. This alignment enables strong spatiotemporal reasoning with minimal training. We also introduce *memory bank sampling*, a frame sampling strategy to extract informative and diverse key frames from sequences, improving efficiency. In the simulated LIBERO environment, *4D-VLA* outperforms existing methods. Additionally, our multi-view simulation dataset, *MV-Bench*, demonstrates superior spatial perception and generalization to novel viewpoints, surpassing current approaches. A limitation of our approach is its reliance on RGB-D input, which introduces hardware restriction.

## Acknowledgments

This work was supported in part by National Natural Science Foundation of China (Grant No. 62376060).

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

# 7 Appendix section

## 7.1 Comparison with ARM4R

Our approach differs from ARM4R in both design and practice. Instead of representation learning with full 3D point clouds and proprioception, we adopt an *end-to-end, action-centric* pretraining pipeline that maps vision–language inputs directly to low-level actions. Inputs are RGB with depth-aligned spatial coordinates (2D patches + $(x, y, z)$ embeddings), keeping the model lightweight and VLM-compatible while avoiding point-cloud encoders.

| Method | Meet-off-grill | Push-buttons | Place-wine | Open-drawer | Avg. |
|---|---|---|---|---|---|
| C2FARM-BC [39] | 20.0 | 72.0 | 18.0 | 20.0 | 32.5 |
| LLARVA [40] | 80.0 | 56.0 | 12.0 | 60.0 | 52.0 |
| PerAct [41] | 84.0 | 48.0 | 12.0 | 80.0 | 56.0 |
| ARM4R [32] | 94.4 | 67.2 | 36.0 | 88.8 | 71.9 |
| *4D-VLA*(Ours) | **95.2** | **79.2** | **45.6** | **89.6** | **77.4** |

Table 6: **RLBench results under the ARM4R protocol.**

**Analysis.** To balance efficiency and spatial coverage, we use our streaming *memory bank sampling* strategy that keeps 5 history frames within a 20-frame context. The current step uses dual views (front + wrist), while history stores only the front view. We also adopt OpenVLA's random-crop augmentation. This configuration provides strong spatial grounding and good runtime, offering a strategy different from ARM4R yet comparably effective on its benchmark.

**Note on supervision.** Our method leverages calibrated intrinsics/extrinsics during training and inference. While such calibration is standard in many robotic datasets, it is an additional source of supervision not explicitly used by ARM4R.

## 7.2 More ablation study

| Action head | FPS | Success rate |
|---|---|---|
| MLP | **12.6** | **29.5** |
| auto regression | 0.6 | 28.1 |
| diffusion | 8.7 | 27.0 |

| Pretrain | Coord. | Prop. | Success rate |
|---|---|---|---|
| ✗ | ✗ | ✗ | 29.5 |
| ✗ | ✗ | ✓ | 16.4 |
| ✗ | ✓ | ✗ | 36.6 |
| ✓ | ✓ | ✗ | **45.0** |

Table 7: **Ablations on heads and inputs (Libero-Long).** Left: action head vs. FPS and success (MLP, autoregressive, diffusion). Right: effect of pretraining, 3D coordinate embedding, and proprioceptive tokens on success.

| Sampling Method | Success ↑ | Latency (ms) ↓ | Peak Mem (MB) ↓ |
|---|---|---|---|
| Single frame | 0.738 | **76.5** | **8342.5** |
| Adaptive Pooling [42] | 0.604 | 150.7 | 8949.1 |
| Grid Pooling [43] | 0.620 | 208.8 | 8852.7 |
| Q-Former (per-frame) | 0.556 | 223.3 | 8812.8 |
| **MBS (Ours)** | **0.866** | 160.0 | 8682.9 |

Table 8: **Frame sampling ablations on Libero-Spatial.** MBS attains the highest success (0.866) with competitive efficiency, while single-frame is fastest and most memory-light but less accurate.

**Action heads.** We conduct additional ablations on LIBERO-LONG. For the action heads (Tab. 7, left), the MLP head yields the highest inference speed with a relatively high success rate. In contrast, the autoregressive head—predicting text tokens for actions—runs slower due to multi-token reasoning.

**Pretrain and spatial components.** Results for other components/operations are shown in Tab. 7 (right). Proprioceptive tokens hurt performance, likely because proprioceptive states (e.g., joint positions, velocities) are highly correlated with action labels, which encourages overfitting to dataset-specific motion patterns rather than learning generalizable visuomotor features.

**Memory bank sampling.** We compare our *memory bank sampling* with three video-style samplers: *Adaptive Pooling* [42], *Grid Pooling* [43], and a *per frame Q-Former*, using InternVL2.5-4B on Libero-Spatial (20 epochs). We report success, latency, and peak memory in Tab. 8. *Adaptive Pooling* caches the full trajectory and uses two thresholds (0.99 for frame selection and 0.96 for spatial compression). We omit its second-stage cross-modal filter due to low language diversity (10 tasks) and observe hyperparameter sensitivity and variable-length tokens that increase memory and reduce accuracy. *Grid Pooling* replaces language tokens with 8 learnable queries and compresses each frame to a $2\times2$ grid (4 tokens), yielding $20\times(4+8)$ historical tokens; it is simple but nonadaptive and can drop key spatial cues. *Q-Former (per frame)* uses 10 learnable queries per frame to extract features but converges slowly and discards too much spatial context, which hurts success. *MBS (ours)* operates online with a window of 20 and a memory of 5. It selects informative key frames from the stream, stores only 5 history frames at $224\times224$, and keeps the current frame at $448\times448$. As a result, MBS achieves the highest success (0.866) with competitive cost (160.0 ms and 8,682.9 MB). Single frame is faster (76.5 ms) but weaker (0.738). The other baselines are slower and less accurate. MBS fits VLA's streaming and causal nature because it avoids full-clip caching, uses relative temporal encoding, and remains compatible with closed-loop control. It yields stronger long horizon performance without extra retraining or heavy compute.

## 7.3 Coordinate system chaos impact analysis

To assess the impact of coordinate system chaos on VLA's performance, we conduct a controlled experiment. The experiment consists of two main steps: First, we deliberately introduce controlled chaos into the LIBERO environment. Then, we compare the performance of models with and without 3D information. Our findings demonstrate that chaotic coordinate transformations significantly degrade model performance while incorporating 3D information effectively alleviates it.

**Chaos generation.** To simulate the diverse viewpoints in the pretraining dataset—where the robot's body is absent from the image, and its coordinate system varies unpredictably—we select trajectories from a specific task in LIBERO-SPATIAL and render each trajectory from 30 distinct viewpoints. For each trajectory and its corresponding viewpoint, we introduce coordinate system chaos by applying a random translation $\mathbf{t} \in \mathbb{R}^3$ and rotation $\mathbf{q} \in \mathbb{SO}(3)$ to the robot's coordinate frame.

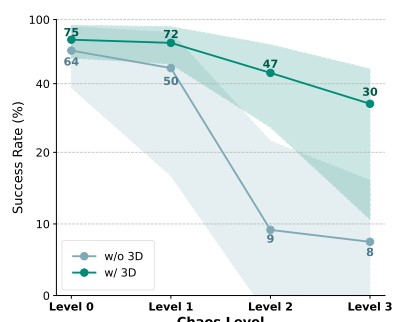

Figure 7: **Success rates under varying coordinate chaos levels.**

After applying the coordinate transformation, the gripper's grasping state in the ground-truth action remains unchanged. However, the rotation offset $\Delta\boldsymbol{\theta}$ and translation $\Delta\boldsymbol{x}$, along with the proprioceptive and camera pose information, undergo the following transformation:

$$\Delta\boldsymbol{\theta}' = \psi^{-1}(\mathbf{q}\psi(\Delta\boldsymbol{\theta})\mathbf{q}^\top), \qquad \Delta\boldsymbol{x}' = \mathbf{q}\Delta\boldsymbol{x},$$
$$\boldsymbol{\theta}' = \psi^{-1}(\mathbf{q}\psi(\boldsymbol{\theta})), \qquad \boldsymbol{x}' = \mathbf{q}\boldsymbol{x} + \mathbf{t},$$
$$\mathbf{R}' = \mathbf{q}\mathbf{R}, \qquad \mathbf{T}' = \mathbf{q}\mathbf{T} + \mathbf{t}. \qquad (5)$$

Here, the function $\psi : \mathbb{R}^3 \to \mathbb{SO}(3)$ maps an Euler angle to its corresponding rotation matrix. The terms $\Delta\boldsymbol{\theta}'$ and $\Delta\boldsymbol{x}'$ denote the transformed action values, while $\boldsymbol{\theta}'$, $\boldsymbol{\theta}$ and $\boldsymbol{x}'$, $\boldsymbol{x}$ represent the transformed and original rotation and position, respectively. Additionally, $[\mathbf{R}'|\mathbf{T}']$ and $[\mathbf{R}|\mathbf{T}]$ correspond to the transformed and original camera poses.

In the subsequent training process, we utilize the transformed action values $\Delta\boldsymbol{\theta}'$ and $\Delta\boldsymbol{x}'$, along with the transformed camera parameters, for model training.

**Implementation details.** We employ a simple baseline, controlling for the involvement of 3D information. The baseline model extracts tokens from a single RGB view, while the alternative model converts an RGB-D frame into spatial vision tokens as input for the LLM Transformer. During testing, we do not apply random rotations or translations to the world coordinate system.

**Experimental results.** We control chaos levels by adjusting the magnitude of random rotations. Level 0 applies no rotation, while levels 1–3 introduce random z-axis rotations of 15°, 30°, and 90°, respectively. Translation $\mathbf{t}$ is randomly set within a range of 0.5. As shown in Fig. 7, without chaos, both models perform well, with 3D information further boosting success rates. Notably, the 3D model shows lower variance across viewpoints. As chaos increases, the non-3D model's performance drops sharply, while the 3D model maintains relatively high success—highlighting the value of 3D cues in handling coordinate system chaos.

