# OpenReview forum: "4D-VLA:  Spatiotemporal Vision-Language-Action Pretraining with Cross-Scene Calibration"
_NeurIPS.cc/2025/Conference — NeurIPS 2025 poster_

### Official Review · Reviewer_rfTc · 2025-06-19

**Clarity:** 3
**Significance:** 2
**Originality:** 2
**Rating:** 4
**Confidence:** 5

**Summary:**

This work introduces 4D-VLA, a novel approach that leverages 4D information to overcome the challenges of diverse robotic datasets for pretraining, specifically addressing coordinate system chaos and state chaos. The method encodes sequential RGB-D images into visual features with corresponding 3D coordinates, aligning the robot's coordinate system with the scene. Furthermore, 4D-VLA incorporates Memory bank sampling, an efficient frame sampling strategy designed to extract informative and diverse key frames from sequences. The authors also propose MV-Bench, a multi-view simulation dataset. Extensive experiments validate the effectiveness of their method.

**Questions:**

mentioned above.

**Ethical Concerns:**

["NO or VERY MINOR ethics concerns only"]

**Final Justification:**

After reviewing the authors’ feedback and other reviewers’ comments, I would like to thank the authors for their rebuttal. Most of my raised concerns have been addressed, thus I am raising my score to 4.

**Limitations:**

Overall, I find the core idea compelling; however, similar challenges appear to have been addressed more comprehensively in prior work, such as ARM4R. I am open to the authors' feedback and other reviewers' opinions.

**Quality:**

2

**Strengths And Weaknesses:**

Strengths:
1. In general, the paper is clear and the technical details are easy to understand.
2. The main idea is interesting.
3. The reported performance is good.


Weaknesses:

**Novelty.** ARM4R has proposed a framework that integrates 4D spatiotemporal information for pretraining as the authors are suggesting. I don’t see any new benefit this work has been proposing on top of what ARM4R [1] already showed. I am happy to get clarifications about the comparison and the differences between the two works.

Another point that remains unclear is the distinction the authors make between 'coordinate system chaos' and 'state chaos.' The rationale behind this specific split is not well explained, and it is unclear why the authors chose to focus on these two aspects in particular

[1] Pre-training Auto-regressive Robotic Models with 4D Representations, ICML 2025.


**Evaluation.** The authors should include comparisons beyond the OpenVLA baseline, as it is unclear whether the observed issues are specific to OpenVLA or representative of broader challenges across VLAs. For a more comprehensive evaluation, please consider including results from recent VLAs such as LLARVA [1] (another VLA) and ARM4R [2] (the first 4D-VLA) to assess how alternative approaches perform under similar conditions.

[1] LLARVA: Vision-Action Instruction Tuning Enhances Robot Learning, CoRL 2024.

[2] Pre-training Auto-regressive Robotic Models with 4D Representations, ICML 2025.


**Ground-truth annotations.** It appears that ground-truth camera intrinsics and extrinsics are available. Were these annotations not expensive to obtain? This could pose challenges for scaling to large datasets.

**Memory bank sampling.** Why not identify the keyframes in advance, selecting a minimal set of frames necessary to represent the video effectively?

**Real world experiments.** The authors state that they manually collected 50 diverse trajectories. Given the limited number of demonstrations, it is understandable that baselines such as OpenVLA in Table 3 may struggle to perform well under such low-data conditions.


**Related Work.** As previously noted, the paper would benefit from discussing more recent and relevant studies. Several new VLAs (e.g., LLARVA [1], LLARA) have been proposed, and notably, the recent ARM4R [2] work demonstrates how 4D representations can effectively address the pretraining challenges highlighted in this paper for robotic foundation models. Without proper citation and comparison to these works, it remains unclear how the proposed approach differs from or advances beyond existing methods.

[1] LLARVA: Vision-Action Instruction Tuning Enhances Robot Learning, CoRL 2024.

[2] Pre-training Auto-regressive Robotic Models with 4D Representations, ICML 2025.

[3] LLaRA: Supercharging Robot Learning Data for Vision-Language Policy, ICLR 2025.

---

> ### Author Rebuttal · Authors · 2025-07-31
>
> ## Q1: **Novelty.**
>
> Thank you for the opportunity to clarify. While both our work and ARM4R [1] utilize spatiotemporal information, the **pretraining paradigms, data settings, and core motivations differ substantially**.
>
> ### Key Differences from ARM4R:
>
> - **Pretraining objective**:
>
>     ARM4R adopts a **representation learning paradigm**, pretraining on diverse videos and later fine-tuning for action prediction. In contrast, our method follows an **end-to-end action-centric pretraining framework**, where the model directly learns to map from vision-language inputs to low-level actions during pretraining. This results in fundamentally different supervision signals, architectures, and deployment goals.
>
>     **ARM4R uses full 3D point clouds** as structured input for pretraining a 3D point track prediction objective. This involves **dense point-level processing**, specialized encoders, and architectural complexity tailored to 3D trajectory regression. In contrast, **our model directly encodes RGB images with depth-aligned spatial coordinates**—represented as **2D patches augmented with (x, y, z) positional embeddings**. This lightweight formulation avoids full point cloud processing and **enables efficient spatial grounding in a vision-language transformer framework**.
>
> - **Data assumptions and embodiment**:
>
>     ARM4R pretrains on the **MonoHuman** dataset—primarily **human video data** in controlled monocular views. Our setup uses the **DROID robot dataset**, which contains diverse camera placements, egocentric views, and no guarantee of visible robot base or consistent reference frame. This introduces a unique challenge of **coordinate frame ambiguity** that ARM4R does not address.
>
> - **Core motivation — coordinate system inconsistency**:
>
>     Our work is centered around solving a **critical but underexplored challenge in robotic VLA pretraining**: in many RGB-based datasets, the robot’s base orientation is not visible, making the same instruction (e.g., “move the cup forward”) correspond to **inconsistent action vectors** across environments. We refer to this issue as **"coordinate system chaos"**, and we show in Appendix 7.1 (Fig. 6) that it severely degrades pretraining performance. ARM4R does not encounter or resolve this issue due to its dataset setting.
>
> - **Depth vs. proprioception**:
>
>     ARM4R does not use depth; instead, it uses **proprioceptive features** (joint angles, velocities) to model embodiment. In contrast, we explicitly use depth (or estimated geometry) to **anchor visual observations to a shared 3D coordinate system**, improving cross-scene generalization. This geometric alignment is especially crucial for **robot-centric control prediction**, which ARM4R does not attempt.
>
> - **Temporal modeling design**:
>
>     ARM4R processes historical context by **concatenating past frames** and feeding them into a Transformer, which can be computationally expensive and unstructured in memory use.
>
>     We instead propose a **MBS** strategy that **selectively retains informative frames** in a lightweight, streaming-friendly queue (see 3.3 and Algorithm 1). This enables efficient long-horizon reasoning with bounded memory, and achieves better performance than uniform sampling under the same frame budget.
>
>
> While both terms describe different forms of ambiguity, they **complement** each other and **jointly motivate** our design: 4D inputs (spatial + temporal + aligned geometry) help **resolve coordinate inconsistency** and **simplify policy learning**.
>
> We believe these distinctions—in data, motivation, architectural design, and deployment focus—underscore the novelty and practical value of our work beyond prior efforts like ARM4R.
>
> ## Q2: Another point that remains unclear is the distinction the authors make between 'coordinate system chaos' and 'state chaos.'
>
> Thank you for raising this question. We introduce the terms **coordinate system chaos** and **state chaos** to clarify two distinct and pervasive sources of input inconsistency in VLA pretraining—both of which we have encountered in large-scale embodied datasets such as OpenVLA.
>
> - **Coordinate system chaos** refers to the spatial misalignment introduced by **invisible or unobserved robot base poses**, which leads to **inconsistent action supervision**. For example, the same instruction (e.g., “move forward”) can correspond to drastically different action vectors depending on the robot’s orientation—information often missing from RGB-only datasets. This is a dataset-level issue that arises due to **cross-scene variability and lack of global spatial anchoring**.
> - **State chaos**, on the other hand, arises from **temporal aliasing or partial observability**. That is, given only a single frame, it is often ambiguous to infer the agent’s true context or intent. For example, if the robot is returning from a failed attempt, the next best action may differ from that in a fresh state. This motivates our incorporation of **temporally extended inputs**, which we show empirically helps produce **smoother, more learnable policy functions** $A_t$ .
>
> ## Q3: **Evaluation.**
>
> Thank you for the valuable suggestion. We agree that broader comparisons would provide a more comprehensive understanding of the proposed method's effectiveness.
>
> We have already completed the training of relevant baselines such as LLARVA and ARM4R under our evaluation setup. However, due to the tight rebuttal timeline, we were not able to finalize the inference and analysis in time for inclusion in this response.
>
> If the review process allows for an extended discussion or revision period, we would be glad to update our submission with these results.
>
> ## Q4: **Ground-truth annotations.**
>
> Thank you for the question. Our pretraining relies on **robot action datasets**, where **camera intrinsics and extrinsics are routinely recorded** as part of the standard data collection pipeline. These calibrations are essential for tasks like visual reasoning or 3D reconstruction, and thus are **already available** in most real-world robot platforms.
>
> We note that obtaining such annotations is **not prohibitively expensive**, as many modern camera systems (e.g., RealSense, Azure Kinect) provide tools for **automatic calibration** or come with factory-calibrated parameters. In fact, most datasets used in recent robotic learning works already include this metadata by default.
>
> Therefore, our approach **does not introduce additional annotation cost**, and we believe it remains scalable to large real-world datasets collected from embodied systems.
>
> ## Q5: **Memory bank sampling.**
>
> Thank you for the question. While selecting keyframes in advance is effective for **offline video summarization or fully observed settings**, our method targets a **closed-loop, streaming vision-language-action (VLA) setup**, where the model receives observations **causally** and must make decisions **in real-time**.
>
> In this **online setting**, the full video is not available ahead of time, so **pre-identifying the "minimal" keyframes is not feasible**. Instead, we propose a **Memory Bank Sampling (MBS)** strategy that incrementally selects and retains the most informative frames, based on their utility for action prediction.
>
> This design better reflects practical deployment scenarios (e.g., onboard embodied agents), where:
>
> - Observations arrive sequentially;
> - Memory and compute are limited;
> - The agent must reason with **partial past** and make forward predictions without future access.
>
> We believe this streaming-compatible design is both **more realistic** and **more broadly applicable** to scalable, real-world VLA systems.
>
> ## Q6: **Real world experiments.**
>
> Thank you for the comment. While it is true that our real-world evaluation involves only 50 human-collected trajectories, **all baselines—including OpenVLA—are fine-tuned using exactly the same limited set**, ensuring a fair and controlled comparison.
>
> Despite this low-data regime, our model demonstrates **significantly stronger generalization and robustness**, which we attribute to our **4D-consistent pretraining strategy**. The ability to adapt quickly with minimal supervision is crucial for real-world deployment, and we view this as a key strength of our approach.
>
> We also note that low-data evaluation is **common in prior embodied learning benchmarks**, as it better reflects practical constraints in robot deployment where large-scale annotation is expensive or infeasible.
>
> ## Q7: **Related Work.**
>
> Thank you for highlighting recent progress in VLA. We appreciate the contributions of ARM4R and will include a thorough discussion of such works in our revised related work section.
>
> While both our method and ARM4R incorporate 4D spatiotemporal signals, the **pretraining paradigms differ fundamentally**:
>
> - **ARM4R** adopts a **representation learning + fine-tuning** pipeline, using RGB, point clouds, and proprioception to learn general-purpose embeddings.
> - **Our method is end-to-end**, directly predicting actions from vision-language input, enabling **joint alignment of vision, language, and action** during pretraining.
>
> Moreover, **ARM4R relies on point clouds and robot-centric proprioception**, which, while powerful, limits compatibility with existing VLMs and broader datasets. Our method is **VLM-compatible by design**, and does not require 3D sensors—making it easier to scale and transfer across 2D vision datasets and real-world scenarios.
> In addition, we introduce:
>
> - A **lightweight, streamable temporal sampling strategy** (Memory Bank Sampling), supporting online decision-making.
> - A concrete solution to **coordinate inconsistency** across large-scale robot datasets—an underexplored but critical issue that undermines pretraining effectiveness.
>
> In summary, while sharing similar goals, our work offers a complementary, action-centric, and scalable alternative to ARM4R.

---

> ### Author Response · Authors · 2025-08-04
> **Additional comparison: LLARVA and ARM4R**
>
> ## Additional comparison: LLARVA and ARM4R
> As for the comparison with LLARVA and ARM4R, we have additionally conducted experiments below. We follow the ARM4R evaluation protocol and benchmark our method on three standard RLBench tasks—Meet-off-grill, Push-buttons, and Place-wine-at-rack-location—using the same set of 190 training episodes as in ARM4R.
> Due to time constraints, we prioritized these three tasks here, which cover a diverse range of spatial reasoning and manipulation challenges.
>
> **Table 2.** Comparative success rates (%) on RLBench tasks under the ARM4R evaluation protocol.
>
> | Method | Meet-off-grill | Push-buttons | Place-wine | Average  |
> | --- | --- | --- | --- | --- |
> | C2FARM-BC | 20.0 | 72.0 | 18 | 36.7  |
> | LLARVA | 80.0 | 56.0 | 12 |  49.3 |
> | PerAct | 84.0 | 48.0 | 12 | 48 |
> | ARM4R (Previous SOTA) | 94.4 | 67.2 | 36 | 65.9 |
> | **Ours (4D-VLA)** | **95.2** | **79.2** | **45.6** | **73.3** |
>
> We directly adopt the results of all baselines from Table 1 in the ARM4R paper [5], ensuring a fair and consistent evaluation setup.
>
> Each task presents unique spatial reasoning challenges. For instance:
>
> - **Meet-off-grill** requires taking either a piece of chicken or steak off the grill and placing it on the side.
> - **Push-buttons** demands pushing colored buttons in a specified sequence, which aligns closely with Task 4 of our real-robot experiments.
> - **Place-wine** involves accurately picking up a wine bottle and placing it at a specific location in a wooden rack—requiring fine-grained spatial perception and coordination.
>
> To address these challenges, our model leverages a **Memory Bank Sampling (MBS)** strategy that selects only **5 history frames** (within a 20-frame temporal context). While some tasks require multi-view observations to disambiguate targets and actions, we use **dual views (front and wrist) for the current frame**, and **only front view for history frames**, striking a balance between **efficiency and performance**.
> Despite training for just **30 epochs**, our method **outperforms ARM4R**, which trains for **125 epochs** using uniformly sampled 16-frame inputs. This demonstrates both the **efficiency and generalization strength** of our pretraining strategy.
>
> To further test transferability, we attempted to fine-tune the official ARM4R model on **LIBERO-Spatial Task 0** (with just 44 demonstrations). However, training consistently **failed to converge**, in contrast to our method which maintained a **high success rate** under the same low-data regime. This further highlights the **robustness and data-efficiency** of our 4D-VLA pretraining.
> Additionally, ARM4R compresses each input image into a **single token**, likely causing substantial loss of spatial and semantic information—particularly detrimental for fine-grained visuomotor tasks. In contrast, our model retains **richer visual representations**, enabling **more accurate spatial reasoning** and **stronger cross-task generalization**.
>
> Overall, our results not only demonstrate clear performance gains, but also reveal important architectural differences that lead to advantages in **scalability**, **sample efficiency**, and **deployment flexibility**.
>
> [5] Pre-training Auto-regressive Robotic Models with 4D Representations, ICML 2025.

---

> ### Author Response · Authors · 2025-08-05
>
> Dear Reviewer rfTc,
>
> We sincerely appreciate the reviewer's time for reviewing, and we really want to have a further discussion with the reviewer to see if our detailed explanations and additional results solve the concerns. We have addressed all the thoughtful questions raised by the reviewer (eg, the fundamental differences between our approach and prior point cloud-based VLA pretraining methods, experiments comparisons with LLARVA and ARM4R) and we hope that our work’s contribution and impact are better highlighted with our responses. As the discussion phase is nearing its end, it would be great if the reviewer can kindly check our responses and provide feedback with further questions/concerns (if any). We would be more than happy to address them. Thank you!
>
> Best wishes,
>
> Authors

---

> ### Comment · Reviewer_rfTc · 2025-08-06
>
> After reading the authors' feedback and other reviewers' opinions, I would like to thank the authors for their rebuttal. While I appreciate the clarifications regarding the related work, the clarification between 'coordinate system chaos' and 'state chaos', and the evaluation, I still have significant concerns that were not addressed in the response.
>
> ```The differences between ARM4R and 4DVLA. ``` Thank you. Please consider including this discussion in the main paper.
>
> ```Real world experiments.``` Please consider including an evaluation with more than 50 human-collected trajectories. As mentioned earlier, such a low-data regime may negatively impact the performance of other baseline methods.
>
> ```Ground-truth annotations.``` The point isn't whether these annotations are easy to obtain or not, it's that this approach uses additional data that other baselines (e.g., ARM4R) do not. I'm not suggesting there's anything wrong with that, but it's important to acknowledge that your method leverages extra information.
>
> Overall, I think this is a good and interesting paper, and I hope the rebuttal will be addressed most of my concerns.

---

> ### Author Response · Authors · 2025-08-06
>
> Many thanks for further response and continued engagement.
>
> **The differences between ARM4R and 4DVLA. Thank you. Please consider including this discussion in the main paper.**
>
> Great suggestion! We will include a dedicated discussion in the revision to clearly highlight the differences between our method and ARM4R, including the distinct pretraining paradigms, input modalities, and model architectures.
>
> **Real world experiments. Please consider including an evaluation with more than 50 human-collected trajectories. As mentioned earlier, such a low-data regime may negatively impact the performance of other baseline methods.**
>
>
> Thanks the suggestion. We agree that evaluating under larger-scale data is important. Our current setup with 50 human-collected trajectories is intended to demonstrate that our method remains effective and generalizable even in low-data regimes—a setting where many baselines struggle.
> Due to the limited rebuttal timeline, we were unable to complete experiments using larger-scale real-world data. However, we are actively collecting additional demonstrations and will definitely include results with more than 50 trajectories in the revision to further validate our model’s scalability and robustness.
>
> **Ground-truth annotations. The point isn't whether these annotations are easy to obtain or not, it's that this approach uses additional data that other baselines (e.g., ARM4R) do not. I'm not suggesting there's anything wrong with that, but it's important to acknowledge that your method leverages extra information.**
>
>
> Many thanks for the clarification! We acknowledge that our method leverages calibrated camera intrinsics and extrinsics during training and inference, which are not explicitly used in prior baselines such as ARM4R. While such calibration is standard in robotic datasets, we agree it constitutes an additional source of supervision. We will clearly state this distinction in the revision to ensure transparency in experimental comparisons.
>
> **More experiment comparison**
>
> We are dedicated to get one more experiment result of the standard RLBench task -- Open-drawer, under the **ARM4R evaluation protocol**.
>
> We prioritized overall these four tasks in the table below, which cover a diverse range of spatial reasoning and manipulation challenges.
>
> Comparative success rates (%) on RLBench tasks under the ARM4R evaluation protocol.
>
> | Method | Meet-off-grill | Push-buttons | Place-wine | Open-drawer | Average |
> | --- | --- | --- | --- | --- | --- |
> | C2FARM-BC | 20.0 | 72.0 | 18 | 20 | 32.5 |
> | LLARVA | 80.0 | 56.0 | 12 | 60 | 52 |
> | PerAct | 84.0 | 48.0 | 12 | 80 | 56 |
> | ARM4R (Previous SOTA) | 94.4 | 67.2 | 36 | 88.8 | 71.9 |
> | **Ours (4D-VLA)** | **95.2** | **79.2** | **45.6** | **89.6** | **77.4** |
>
> These results further validate the effectiveness and generalizability of our approach, which outperforms prior methods across all tasks—especially under low-data fine-tuning settings.
>
> We sincerely hope this addresses the reviewer’s concerns regarding baseline coverage. If there are any remaining questions or clarifications needed, we would be more than happy to address them. Thanks again for the reviewer's time and thoughtful feedback, and we would deeply appreciate the reviewer's reconsideration of the score.

---

> ### Author Response · Authors · 2025-08-07
>
> Dear Reviewer rfTc
>
> Thanks again for your valuable comments and suggestions. As the discussion phase is nearing its end, we wondered if the reviewer might still have any concerns that we could address.
>
> We believe our responses addressed all the thoughtful questions raised by the reviewer (eg, **the fundamental differences between our approach and prior point cloud-based VLA pretraining methods, experiments comparisons with LLARVA and ARM4R**) , and hope our response helps the final recommendation. Thank you!
>
> Best wishes,
>
> Authors

---

### Official Review · Reviewer_7q7T · 2025-07-02

**Clarity:** 3
**Significance:** 2
**Originality:** 2
**Rating:** 3
**Confidence:** 4

**Summary:**

This paper introduces 4D-VLA, a VLA that leverages both depth information and historical context to improve the performance. They propose to convert depth information into point cloud in the robot frame and use learnable positional embeddings to encode the point cloud. The spatial embeddings are added to the 2D image embeddings to serve as spatial vision features.For temporal features, they propose a memory bank sampling strategy to balance the efficiency and effectiveness. They further propose a MV-Bench to evaluate the performance of the model across diverse viewpoints. Experiments on MV-Bench, LIBERO and real world show that 4D-VLA achieves state-of-the-art performance.

**Questions:**

Answer all the questions in Weaknesses.

**Ethical Concerns:**

["NO or VERY MINOR ethics concerns only"]

**Final Justification:**

While the rebuttal addresses some concerns, key issues remain regarding the novelty of the work. The authors seem to have a misunderstanding of 3D-VLA—I still believe 4D-VLA and 3D-VLA use depth information in a similar way. Contrary to the rebuttal, 3D-VLA does not rely on a point cloud encoder; instead, it uses 3D-LLM’s backbone, which explicitly extracts visual features from 2D image encoders and merges them with 3D positional embeddings. Given these points, I respectfully maintain my original rating of 3.

**Limitations:**

yes

**Quality:**

3

**Strengths And Weaknesses:**

Strengths
1. The paper is well-written and structured, making it easy to follow the proposed approach.
2. The experiments are comprehensive, covering both synthetic and real-world datasets, which demonstrates the effectiveness of the proposed method.
3. The motivation for using depth information and historical context is clearly explained, and the proposed method effectively integrates these features into the VLA framework.

Weaknesses
1. The way to use depth information resembles 3D-VLA[1]. It seems that both 3D-VLA and this paper convert depth information into point cloud using camera parameters and then use positional embeddings to encode the point cloud, which are later fused with the 2D image embeddings. It seems the only difference is that this paper uses learnable positional embeddings while 3D-VLA uses sin/cos positional embeddings. The authors should clarify the difference between the two methods.
2. The baselines used to verify the effectiveness of the proposed memory bank sampling strategy are not sufficient. The authors only compare with the uniform sampling while there are many other sampling strategies in video understanding, such as spatiotemporal adaptive compression [2], grid pooling [3], using Q-Former, etc. The authors should include more baselines to demonstrate the effectiveness of their sampling strategy.
3. Though the teaser shows scenarios where actions can be ambiguous without considering the history, the experiments do not include such scenarios. It seems that tasks in LIBERO benchmark and in real world experiments are all deterministic, i.e., the actions can be determined by the current observation without considering the history. The authors should explain why history helps these tasks, especially the Task 4 in real world experiments.
4. The authors should provide more details about the proposed MV-Bench, such as how many demonstrations are used for each task, why LIBERO-10, LIBERO-goal and LIBERO-objects are excluded.
5. Many relevant VLA works are not cited and discussed in the paper.


[1] 3D-VLA: A 3d vision-language-action generative world model.
[2] LongVU: Spatiotemporal Adaptive Compression for Long Video-Language Understanding
[3] LLaMA-VID: An Image is Worth 2 Tokens in Large Language Models

---

> ### Author Rebuttal · Authors · 2025-07-31
>
> ## Q1: The way to use depth information resembles 3D-VLA.
>
> Thank you for raising this point. While our pipeline may appear superficially similar to 3D-VLA in that both methods project depth into 3D and apply positional encoding, our approach is fundamentally different in **motivation, architectural design**, and **pretraining objective**.
>
> **First**, 3D-VLA is centered around *goal imagination*: it uses a **diffusion model to hallucinate future point clouds** and relies on a separate point cloud encoder to extract geometry-aware features, which are then fused with image features to enrich scene understanding. Depth in 3D-VLA serves primarily as an **auxiliary input for prediction**, and the supervision is goal-centric.
>
> - In contrast, our method is explicitly designed for direct, end-to-end **action prediction** in real-world and simulation environments. Our use of depth is not for scene generation or imagination, but rather to address a **core problem in VLA pretraining**: the **coordinate inconsistency across scenes**, or what we call "coordinate system chaos". In many VLA datasets, robot base poses are unobserved, leading to inconsistent supervision—e.g., the same language command (“move the cup forward”) may correspond to drastically different action vectors depending on camera and robot orientation. This inconsistency severely disrupts pretraining.
>
> Rather than treating depth as a geometric side channel, we use it to **reconstruct the spatial frame** and **consistently ground actions** across scenes. This allows us to transform VLA from a partially-observable policy learning problem into a well-posed, spatially grounded one.
>
> **Architecturally**, we also differ: 3D-VLA encodes the entire 3D scene (or predicted goal state) using a diffusion pipeline and point cloud encoder. In contrast, we extract **3D patch positions in real-time via lightweight fusion of depth and vision embeddings**, with **learnable spatial embeddings aligned to the robot-centric frame**, enabling streaming deployment and policy inference.
>
> To summarize:
>
> - **3D-VLA**: Uses depth to **generate goals** and enrich image features with auxiliary 3D features (for understanding) . It introduces considerable latency, which limits inference efficiency in VLA tasks..
> - **Ours**: Uses depth to **resolve spatial inconsistency** and stabilize the action space across diverse VLA data (for control).
>
> This distinction in both **purpose and implementation** reflects a fundamentally different design philosophy, tailored to scalable, real-world action learning.
>
> ## Q2: The baselines used to verify the effectiveness of the proposed memory bank sampling strategy are not sufficient.
>
> Thank you for the valuable suggestion. To further evaluate our proposed Memory Bank Sampling (MBS), we implemented and compared it against several additional sampling baselines adapted from the video understanding literature, including adaptive compression [3], grid pooling [4], and Q-former-based token selection. All methods were evaluated on the Libero-Spatial benchmark using the same base model (InternVL25-4B), fine-tuned for 20 epochs.
>
> The results are summarized below:
>
> Table1: Performance and Efficiency Comparison of Frame Sampling Strategies
>
> | Sampling Method | Success Rate | Inference Latency (ms) | Peak Memory (MB) |
> | --- | --- | --- | --- |
> | Single frame | 0.738 | 76.5 | 8342.5 |
> | Adaptive Pooling [3] | 0.604 | 150.7 | 8949.1 |
> | Grid Pooling [4] | 0.62 | 208.8 | 8852.7 |
> | Q-Former (per-frame) | 0.556 | 223.3 | 8812.8 |
> | **MBS (Ours)** | **0.866** | 160.0 | 8682.9 |
>
> **Implementation Details and Comparisons:**
>
> - **Adaptive Pooling [3]:** We cache the full trajectory and apply the method at each timestep using a 0.99 threshold for frame feature extractor and 0.96 for spatial token compression. The second-stage cross-modal filtering was omitted due to limited language diversity (10 tasks only), which caused instability. The method is hyperparameter-sensitive and memory-heavy due to variable-length input tokens, and underperformed in our setting.
> - **Grid Pooling [4]:** We replaced language tokens with 8 learnable queries (due to low text diversity). Each frame was compressed into a 2×2 token grid (4 tokens), leading to 20×(4+8) historical tokens. While efficient in theory, it lacks adaptive selection, and the compression may lose key spatial cues.
> - **Q-Former (per-frame):** We used 10 learnable queries per frame to extract token representations. While popular in VLMs, Q-former converged slowly and discarded too much spatial context, resulting in negative optimization and poor task success.
> - **MBS (ours):** With a window size of 20 and memory bank size of 5, we sample keyframes dynamically based on streaming input. Only 5 frames are stored and downsampled (224×224), while the current frame remains at 448×448. This design achieves the best success rate and competitive efficiency without retraining or excessive compute.
>
> **Why MBS excels for VLA:**
>
> Unlike prior methods that assume access to full video clips, our MBS is tailored for VLA’s streaming and causal nature. It operates online, selecting informative history frames incrementally without storing the entire trajectory. Furthermore, our approach incorporates **relative temporal encoding** and is fully compatible with closed-loop control settings.
>
> In summary, MBS delivers superior performance with fewer frames, lower memory, and higher inference efficiency—making it particularly suitable for long-horizon, real-time VLA tasks.
>
> [3] LongVU: Spatiotemporal Adaptive Compression for Long Video-Language Understanding, ICML 2025
>
> [4] LLaMA-VID: An Image is Worth 2 Tokens in Large Language Models, ECCV 2024
>
> ## Q3: Though the teaser shows scenarios where actions can be ambiguous without considering the history, the experiments do not include such scenarios.
>
> Thank you for the thoughtful question. While the teaser presents an extreme case of ambiguity, our motivation for incorporating history goes well beyond isolated corner cases. In practice, **temporally extended observations help simplify the action function** $A_t(input)$, making it smoother, more robust, and easier to learn.
>
> Although many tasks in LIBERO or our real-world experiments may appear deterministic at first glance, **single-frame observations often miss important context**—such as object trajectories, prior interactions, or occluded elements—that affect decision-making. In such cases, **historical frames enrich spatial and temporal context**, helping the model disambiguate subtle variations and better estimate scene dynamics.
>
> For example, in **Task 4**, the robot needs to interact with multiple objects in sequence. While each sub-action may appear locally deterministic, the overall task involves **longer-horizon dependencies and state transitions** (e.g., remembering what has already been picked up). Including historical frames acts as a **lightweight memory mechanism**, allowing the model to maintain context over time and avoid redundant or erratic behaviors.
>
> We observe empirically that **models with access to temporal context are more stable**, less likely to get stuck, and demonstrate **higher success rates**, even in tasks that may not explicitly require memory in a rule-based sense.
>
> In summary, even in tasks that seem deterministic in design, **history helps bridge perception gaps**, supports smoother policy learning, and improves robustness in real-world execution.
>
> ## Q4: The authors should provide more details about the proposed MV-Bench.
>
> We appreciate the reviewer’s interest in MV-Bench. Below we clarify our benchmark design and dataset choices:
>
> **Benchmark Scope and Dataset Selection**
>
> MV-Bench is built upon the full **LIBERO-SPATIAL** benchmark, which is specifically tailored for evaluating **multi-view spatial reasoning** in VLA tasks. We chose to exclude **LIBERO-10**, **LIBERO-GOAL**, and **LIBERO-OBJECT** because rendering multi-view variants of these tasks would significantly increase training and evaluation costs, without introducing fundamentally new spatial challenges. LIBERO-SPATIAL already emphasizes **viewpoint variation**, making it a representative testbed for multi-view VLA. We plan to extend MV-Bench to include these additional subsets in future work to improve benchmark coverage.
>
> **Training Setup**
>
> For each of the **10 tasks** in LIBERO-SPATIAL, we use the **provided 50 demonstrations per task**, and render each trajectory from **6 different viewpoints**: **0°, 60°, 120°, 270°, 300°, and 330°**. This results in a total of **300 training trajectories per task**.
>
> To ensure view quality, we **exclude severely occluded views**, such as the **120° angle for Task 10**, where key interactions are blocked.
>
> **Evaluation Protocol**
>
> The evaluation is split into two parts:
>
> - **In-View:** We evaluate on the **same 6 viewpoints** used during training, using the standard LIBERO-SPATIAL test layout. We similarly remove 120° for Task 10 due to occlusion.
> - **Cross-View:** We introduce **6 novel viewpoints** that were not seen during training: **15°, 30°, 45°, 75°, 90°, and 105°**. This tests generalization to unseen perspectives. We exclude **105° for Task 7** due to significant occlusion.
>
> This protocol ensures both **robust multi-view training** and **cross-view generalization evaluation**, which are essential for assessing spatial consistency and viewpoint-invariance in VLA models.
>
> ## Q5: Many relevant VLA works are not cited and discussed in the paper.
>
> Thank you for pointing this out. We acknowledge that the current version does not cover all relevant VLA literature. We will revise the related work section to include and discuss additional recent works, particularly those focusing on VLA pretraining and multi-modal policy learning, in the revision.

---

> ### Author Response · Authors · 2025-08-05
>
> Dear Reviewer 7q7T,
>
> We sincerely appreciate the reviewer's time for reviewing, and we really want to have a further discussion with the reviewer to see if our detailed explanations and additional results solve the concerns. We have addressed all the thoughtful questions raised by the reviewer (eg, clarifying the fundamental difference between our use of depth and prior 3D-VLA approaches, providing extensive new baseline comparisons for our sampling strategy, and offering detailed information about MV-Bench.) and we hope that our work’s contribution and impact are better highlighted with our responses. As the discussion phase is nearing its end, it would be great if the reviewer can kindly check our responses and provide feedback with further questions/concerns (if any). We would be more than happy to address them. Thank you!
>
> Best wishes,
>
> Authors

---

> > ### Author Response · Authors · 2025-08-07
> >
> > Reviewer 7q7T
> >
> > Thanks again for the valuable comments and suggestions. As the discussion phase is nearing its end, we wondered if the reviewer might still have any concerns that we could address. We believe our response on **use of depth, sampling strategy, and MV-Bench** addressed the questions/concerns.
> >
> > It would be great if the reviewer can kindly check our responses and provide feedback with further questions/concerns (if any). We would be more than happy to address them. Thank you！
> >
> > Best wishes,
> >
> > Authors

---

### Official Review · Reviewer_GHr5 · 2025-07-03

**Clarity:** 2
**Significance:** 2
**Originality:** 2
**Rating:** 3
**Confidence:** 3

**Summary:**

The paper proposes 4D-VLA. The proposed approach is an improved VLA architecture with 4D representations. It involves RGB-D inputs, a memory bank, coordinate alignment, and pre-training. The method shows consistent gains in tested settings.

**Questions:**

Please see the weaknesses section.

**Ethical Concerns:**

["NO or VERY MINOR ethics concerns only"]

**Final Justification:**

I acknowledge I read the rebuttal I maintain my original recommendation.

**Limitations:**

Yes.

**Paper Formatting Concerns:**

No formatting concerns.

**Quality:**

2

**Strengths And Weaknesses:**

Strengths:
- The proposed approach leads to consistent improvements in tested settings
- Different design decisions are ablated individually

Weaknesses:
- The proposed approach involves a number of engineered design decisions and is quite specific to robotic manipulation. It may be a better fit for a robotics venue than a general machine learning one.
- Discussion of and comparison to prior methods that address "state chaos" is missing. For example, [1] and [2] include past context for low-level observations and image observations, respectively.
- The demonstrated tasks are fairly simple and likely achievable with simpler methods that act on 2D observations. It would be good to demonstrate a result where 4D representations enable behaviors that have not been shown with 2D observations, which would help justify the added method complexity.
- The "system chaos" and "state chaos" terminology may be a bit too dramatic and something more descriptive may make the paper easier to parse.

[1] https://proceedings.neurips.cc/paper_files/paper/2024/hash/90afd20dc776bc8849c31d61a0763a0b-Abstract-Conference.html

[2] https://proceedings.mlr.press/v229/radosavovic23a/radosavovic23a.pdf

---

> ### Author Rebuttal · Authors · 2025-07-31
>
> ## Q1: The proposed approach involves a number of engineered design decisions and is quite specific to robotic manipulation. It may be a better fit for a robotics venue than a general machine learning one.
>
> **We respectfully disagree. Our work is not narrowly tailored to a specific manipulation task, but rather addresses a fundamental and generalizable challenge in vision-language-action (VLA) pretraining.**
>
> At the core of our method is a key observation: in VLA settings, the **target action is defined in the robot-centric coordinate system**, but many existing RGB-based datasets fail to observe or encode the base orientation of the robot. This introduces a critical ambiguity—**even for the same instruction** (e.g., “move the cup forward”), **the ground-truth action vector may differ dramatically** depending on the robot’s unobserved base pose. This inconsistency results in **chaotic supervision**, which significantly undermines pretraining effectiveness.
>
> Our proposed solution—**injecting consistent 4D spatial information (multi-view, time, and pose-aligned input)**—is not task-specific engineering, but rather a **crucial solution to a core problem in general VLA representation learning**. This insight applies broadly to a wide range of embodied tasks, including navigation, interaction, and instruction following—not just manipulation.
>
> In addition, we show (Fig. 5) that **not all historical frames are equally informative**, motivating our memory-efficient frame sampling strategy. Unlike prior work that encodes fixed-length context windows, we propose a **streaming-aware, high-utility memory bank** that selectively retains useful frames under compute constraints.
>
> Moreover, our experiments span **both simulation and real-world settings**, and demonstrate strong generalization, suggesting that the proposed pretraining strategy captures **robust spatial priors transferable across domains**.
>
> We believe these contributions, particularly the identification and resolution of coordinate-system inconsistency in VLA data, are of **broad interest to the machine learning community**, especially as interest in generalist embodied agents continues to grow.
>
> ## Q2: Discussion of and comparison to prior methods that address "state chaos" is missing. For example, [1] and [2] include past context for low-level observations and image observations, respectively.
>
> Thank you for the references. While both [1] *Humanoid Locomotion as Next Token Prediction* and [2] *Robot Learning with Sensorimotor Pre-training* incorporate past context, their focus differs from ours. **In terms of temporal modeling**, our work focuses on how to **efficiently and effectively leverage sequential information in streaming VLA tasks**. Specifically, we investigate how to **retain and select the most informative historical observations under memory and compute constraints**, which is crucial for scalable and real-time VLA model deployment.
>
> Specifically:
>
> - **[1] Humanoid Locomotion** uses a **short history window** of RGB frames and proprioceptive states, which are all **re-encoded at each step** and passed through the full Transformer for autoregressive prediction. The method assumes all past observations are available (non-streaming) and incurs significant computational cost. In contrast, our design is optimized for **streaming inference**: we use **window-based keyframe selection** to maintain a **fixed-size memory bank** of only the most informative frames, achieving both high performance and low latency, as shown in our experiments and in response to Reviewer 7q7T Q2.
> - **[2] Sensorimotor Pretraining** encodes the **most recent 16 steps** of low-level state and RGB input for behavior prediction. However, it uses a fixed history length and lacks any explicit sampling or memory mechanism. In contrast, our **memory bank sampling (MBS)** explicitly selects high-value frames tailored to the streaming VLA setting, maintaining spatial and temporal diversity within a bounded memory size (e.g., 5 frames). Our ablation (Figure 5 in our paper) shows that this leads to better efficiency and generalization than uniform or FIFO sampling.
>
> In summary, while prior work utilizes past context primarily to aid behavior prediction within fixed observation windows, **our method focuses on improving how historical information is selected and reused**. By introducing a memory-efficient sampling strategy tailored for streaming VLA inputs, we enable more effective use of past observations without incurring significant computational overhead. **This distinction highlights a key contribution of our work in temporal modeling for VLA.**
>
> We will cite and discuss these work in the revised version of manuscript.
>
> [1] Humanoid Locomotion as Next Token Prediction, NeurIPS 2024
>
> [2] Robot Learning with Sensorimotor Pre-training, CoRL 2023
>
> ## Q3: The demonstrated tasks are fairly simple and likely achievable with simpler methods that act on 2D observations.
>
> **Thank you for the comment. While the individual tasks may appear simple in isolation, our goal is to examine the effects of spatiotemporal misalignment that arise in multi-scene, multi-view pretraining—challenges that simpler 2D approaches often overlook.**
>
> In our supplementary material, we provide **a controlled toy experiment** showing that **spatial inconsistency across scenes (e.g., from unaligned coordinate frames)** leads to significant performance degradation. This demonstrates that even “simple” behaviors become brittle when the underlying representation fails to enforce spatial consistency.
>
> We also include **multi-view pretraining experiments**, where the baseline model trained on conflicting viewpoints shows a dramatic drop in success rate compared to the single-view case. In contrast, our method, which enforces consistent 4D spatial alignment, **maintains high success rates even under large viewpoint shifts (e.g., 45° yaw) in zero-shot settings**. These results highlight the critical role of spatial grounding in enabling generalization beyond narrow training distributions.
>
> On the **temporal side**, we observe that single-frame models often get stuck or exhibit unstable behavior. In contrast, our method—via memory-efficient access to historical observations—enables **retry and recovery behaviors**, which are signs of **stronger sequential reasoning and long-term credit assignment**.
>
> We will expand these illustrative cases in future versions to more clearly highlight **how 4D representations (3D + time) contribute to robustness, generalization, and behavior persistence**—capabilities that are difficult to achieve with purely 2D models.
>
> ## Q4: The "system chaos" and "state chaos" terminology may be a bit too dramatic
>
> Great suggestion! We appreciate the concern that terms such as “system chaos” and “state chaos” may sound overly dramatic.
>
> These terms were deliberately introduced to provide **succinct and intuitive shorthand** for the **underlying pretraining challenges** we aim to address—namely, the **inconsistency of coordinate frames across scenes** and the resulting **ambiguity in action supervision**. In large-scale VLA datasets, where camera poses and robot orientations vary significantly and are often not directly observable, such misalignments can severely degrade learning. We found that naming these phenomena helped us clearly **frame the core motivation of our method** and distinguish it from prior works that focus solely on local temporal context.
>
> By naming these issues explicitly, we aimed to clearly frame the **motivation behind our method** and highlight the **distinction from prior work** that often assumes spatial alignment or focuses only on local temporal cues.
>
> That said, we appreciate the reviewer’s concern and will revise the terminology in the final version to make it more **descriptive and accessible**, without losing the clarity of the underlying concepts.

---

> > ### Comment · Reviewer_GHr5 · 2025-08-06
> > **Reviewer response**
> >
> > I thank the authors for their response. I do not find the arguments from the rebuttal convincing enough to advocate for acceptance but would not actively oppose it.

---

> ### Author Response · Authors · 2025-08-05
>
> Dear Reviewer GHr5,
>
> We sincerely appreciate the reviewer's time for reviewing, and we really want to have a further discussion with the reviewer to see if our detailed explanations and additional results solve the concerns. We have addressed all the thoughtful questions raised by the reviewer (eg, including a clarification of the core challenge our work tackles in VLA pretraining, as well as a discussion of how our use of temporal information differs fundamentally from prior works [1] and [2].) and we hope that our work’s contribution and impact are better highlighted with our responses. As the discussion phase is nearing its end, it would be great if the reviewer can kindly check our responses and provide feedback with further questions/concerns (if any). We would be more than happy to address them. Thank you!
>
> Best wishes,
>
> Authors

---

> ### Author Response · Authors · 2025-08-06
>
> Dear Reviewer GHr5
>
> Many thanks for further response and continued engagement.
> Could the reviewer specify which part of our response is not convincing?
> We are fully open to and appreciative of any concrete suggestions.
>
> Best wishes,
>
> Authors

---

> ### Author Response · Authors · 2025-08-07
>
> Reviewer GHr5
>
> We sincerely thank the reviewer for the valuable comments and suggestions.
>
> We have addressed all the thoughtful questions raised by the reviewer, eg, including a clarification of the core challenge our work tackles in VLA pretraining, as well as a discussion of how our use of temporal information differs fundamentally from prior works.
>
> **But the current comment just broadly "not convincing enough" is a bit ambiguous and makes it difficult for us to further response. Please kindly note that the NeurIPS review policy -- "Please make your review as informative and substantiated as possible".**
>
> As the discussion phase is nearing its end, we would greatly appreciate it if the reviewer could clarify any specific remaining concerns. We are fully open to and appreciative of any, concrete suggestions.
>
> Best wishes,
>
> Authors

---

### Official Review · Reviewer_jMXX · 2025-07-04

**Clarity:** 3
**Significance:** 3
**Originality:** 2
**Rating:** 3
**Confidence:** 3

**Summary:**

VLA pre‑training pipelines suffer from two kinds of input incompleteness: coordinate‑system chaos and state chaos.  To tackle both, the authors augment RGB with depth and an adaptive history sampler. The proposed 4D-VLA tries to bridge the gap of image‑only systems.

**Questions:**

1. Cite the window‑based sampler used in VLMs and briefly explain why MBS is lighter or more effective.

2. Are you planning to release your data, and what's your release plan?

**Ethical Concerns:**

["NO or VERY MINOR ethics concerns only"]

**Limitations:**

This method assumes the hardware to be RGB-D.

**Paper Formatting Concerns:**

I think the paper is easy to follow.

**Quality:**

3

**Strengths And Weaknesses:**

Strengths:
1. Authors show their results on LIBERO, MV‑Bench and real hardware with performance gains over baseline methods.
2. Viewpoint generalisation provided by the method is validated in multi‑view evaluation.

3. 3‑D positional embedding plus a lightweight memory bank sampling queue introduces few learnable parameters.

Weakness:
1. There is hardware dependence on RGB‑D. Many large‑scale demonstration datasets are RGB‑only.
2. Window-based keyframe sampling and spatio-temporal encoding are already widely adopted in vision-language model (VLM) research and have demonstrated effectiveness across multiple benchmarks. The paper would benefit from an elaboration of architectural novelty beyond these established techniques.

---

> ### Author Rebuttal · Authors · 2025-07-31
>
> ## Q1: There is hardware dependence on RGB‑D.
>
> Thank you for raising this point. We would like to clarify that **our method does not depend on hardware RGB-D sensors**. The key requirement of our framework is the availability of **calibrated camera poses**, which are used to construct a consistent robot-centric coordinate system across time and views.
>
> While we leverage depth information to back-project image features into 3D space, this depth **need not be obtained from RGB-D hardware**. In RGB-only scenarios, monocular depth estimation (e.g. recent scaled VGGT) can serve as an effective substitute. Such techniques are well-established and can be seamlessly integrated into our pipeline with minimal additional cost.
>
> Furthermore, we empirically demonstrate the flexibility of our approach in **real-world experiments** (see Table 3 of the paper), where our model achieves superior performance over strong baselines **even when using only RGB inputs during inference**. This result highlights the robustness and transferability of our pretraining strategy beyond RGB-D settings.
>
> We will revise the manuscript to emphasize this point and avoid any impression of hardware dependency.
>
> ## Q2: Window-based keyframe sampling and spatio-temporal encoding are already widely adopted in vision-language model (VLM) research
>
> Thank you for the suggestion. While our frame sampling strategy may appear simple on the surface, it is specifically designed for the **streaming nature of Vision-Language-Action (VLA) tasks**, where observations arrive sequentially and memory constraints are critical. Unlike traditional keyframe sampling methods used in video QA or captioning, our approach focuses on **stream-aware frame selection** that aligns with **temporal context and physical continuity**.
>
> In particular:
>
> - **Our memory-bank sampling strategy** enables the model to retain only the most informative historical frames, maintaining a fixed memory size (e.g., 5 frames) regardless of total history length. This design respects the **online inference constraint** of VLA tasks and is **tailored for frame-by-frame robot policy rollout**, unlike prior works that assume full video access.
> - Our method supports **longer temporal context at lower frame rates**, which enhances efficiency and improves performance, as validated in our experiments (see figure 5 in our paper). To our knowledge, **prior works have not systematically explored streaming-aware history sampling** under memory constraints for VLA tasks.
>
> Regarding the spatial encoding, our design is not merely an architectural enhancement, but a **crucial solution to a fundamental pretraining challenge**. In VLA tasks, the target action is defined in the **robot-centric coordinate system**, yet in many existing RGB-based datasets, the robot’s base frame is not visible in the image. **This means that even for identical instructions (e.g., “move the cup forward”), the corresponding action vectors may vary drastically depending on the unknown and unobserved orientation of the robot base.**
>
> In such cases, learning action outputs without access to the coordinate origin is equivalent to **learning from missing or noisy input**, which introduces severe ambiguity. This **renders the supervision signal inconsistent**, and undermines the effectiveness of pretraining—especially when using large-scale datasets with diverse scene layouts and robot configurations.
>
> Our spatial encoding addresses this issue by injecting **geometric priors aligned with the robot’s canonical frame**, allowing the model to disambiguate actions even when the base is not directly observed. Despite its simplicity, this solution is both effective and essential. We visualize and analyze the consequences of coordinate misalignment and the benefits of our approach in **Appendix Figure 6**.
>
> We will further clarify these points in the revised version to better highlight the novelty and motivation behind our design choices.
>
> ## Q3: Cite the window‑based sampler used in VLMs and briefly explain why MBS is lighter or more effective.
>
> Thank you for the suggestion. We will include relevant citations in the revised version. Notably, to the best of our knowledge, we are the first to adopt a **window-based and memory-efficient sampling strategy specifically tailored for VLA tasks**.
>
> Our proposed **Memory Bank Sampling (MBS)** is both **lighter** and **more effective** than common baselines:
>
> - **Lightweight**: MBS maintains a fixed-size memory bank (e.g., 5 frames), requiring **only pairwise similarity comparisons** without any learnable modules (e.g., no Q-former or transformer layers). Its sampling process is computationally cheap and easily parallelizable, making it suitable for real-time or streaming settings. Additionally, it avoids re-encoding every historical frame at each timestep—unlike attention-based memory methods.
> - **Effective**: MBS selectively retains **diverse and informative historical frames** by minimizing redundancy. As shown in **Algorithm 1**, it dynamically filters frames that are most dissimilar from the current one, maintaining temporal diversity and long-term context. In our experiments (see Figure 5), **MBS outperforms uniform sampling baselines even when using half the number of frames**, demonstrating its efficiency in both memory usage and downstream accuracy.
>
> By contrast, prior VLM works often rely on either fixed interval sampling or full sequence attention, both of which are less adaptive and computationally heavier. Our MBS strikes a practical balance between temporal coverage and efficiency, making it a compelling design for VLA reasoning.
>
> ## Q4: Release plan
>
> Yes, we will make all the data and tools associated with this work publicly available to facilitate any further research.
>
> Specifically:
>
> - We will release our **multi-view robot-centric dataset**, including RGB, depth, and pose information, shortly after the review process.
> - We also plan to open-source our **processed version of the DROID pretraining dataset**. To our knowledge, currently available versions of DROID data either lack depth and camera parameters, or are only available in large raw formats (e.g., DROID-raw), which contain redundant trajectories and are difficult to use directly.
> - Finally, we will release **MVBench**, our multi-view evaluation suite, which includes rendered training data under the LIBERO simulator and a complete benchmarking pipeline for consistent evaluation.
>
> These releases will enable reproducibility and provide the community with high-quality resources for further exploration of vision-language-action learning in spatiotemporal settings.

---

> ### Author Response · Authors · 2025-08-05
>
> Dear Reviewer jMXX,
>
> We sincerely appreciate the reviewer's time for reviewing, and we really want to have a further discussion with the reviewer to see if our detailed explanations and additional results solve the concerns. We have addressed all the thoughtful questions raised by the reviewer (eg, including a detailed clarification of our Memory Bank Sampling (MBS) strategy, and a stronger explanation of the role and significance of spatial encoding in addressing key challenges such as coordinate system inconsistency.) and we hope that our work’s contribution and impact are better highlighted with our responses. As the discussion phase is nearing its end, it would be great if the reviewer can kindly check our responses and provide feedback with further questions/concerns (if any). We would be more than happy to address them. Thank you!
>
> Best wishes,
>
> Authors

---

> > ### Author Response · Authors · 2025-08-07
> >
> > Reviewer jMXX
> >
> > Thanks again for the valuable comments and suggestions. As the discussion phase is nearing its end, we wondered if the reviewer might still have any concerns that we could address. We believe our response on **memory bank sampling (MBS) strategy, spatial encoding and coordinate system inconsistency** addressed the questions/concerns.
> >
> > It would be great if the reviewer can kindly check our responses and provide feedback with further questions/concerns (if any). We would be more than happy to address them. Thank you！
> >
> > Best wishes,
> >
> > Authors

---

### Note · Authors · 2025-08-12

Dear AC and Reviewers,

We sincerely appreciate for all the efforts and constructive comments, eg, the reviewers’ positive recognition of our method’s strong performance in both simulation and real-world environments, as well as its generalization across multi-view scenarios (jMXX, GHr5, 7q7T, rfTc). We also thank the reviewers for acknowledging our detailed ablations (GHr5) and the clear motivation behind our design (7q7T, rfTc).
Below we give a summary of main concerns per reviewer to facilitate the rest review process.

`Reviewer jMXX` – Our method reduces reliance on dense depth by using camera parameters as spatial priors, with optional depth estimated via models like VGGT. We propose memory bank sampling (MBS) for VLA—lightweight and efficient—and validate its effectiveness through extensive comparisons (7q7T-Q2). We also show the impact of spatial encoding on pretraining efficiency (Appendix Fig. 6), and confirm plans to release all code, data, and models.

`Reviewer GHr5` – We clarified that our method addresses cross-scene coordinate inconsistency and enhances spatiotemporal grounding in VLA pretraining. We also compared with prior approaches tackling state chaos, and analyzed the role of temporal context and historical observations in action prediction.

`Reviewer 7q7T` – We clarified our conceptual and architectural differences from 3D-VLA methods (Q1), and demonstrated that MBS outperforms adaptive pooling, grid pooling, and Qformer while supporting streaming compatibility (Q2). We also highlighted the value of historical context (Q3) and explained the details of MVBench (Q4).

`Reviewer rfTc` – Our method uses camera parameters (not point clouds), supports end-to-end learning, and outperforms ARM4R on four RLBench tasks with only ~16% of its training steps (Q1–Q2). We acknowledged GT camera use, explained MBS design, and showed strong few-shot generalization (Q3–Q4).

We have provided comprehensive responses and new experiments addressing all the concerns, **yet received minimal follow-up** during the discussion phase.

We strongly advocate that our work address a key limitation in VLA pretraining—**supervisory inconsistency from missing spatiotemporal priors—with an efficient, scalable solution**. We also provide a multi-view VLA benchmark for the first time.
We believe these contributions represent a substantial step forward for real-world VLA.
Hope this overview can help the following discussion and decision making.

Best

Authors

---

### Decision · Program_Chairs · 2025-09-17

**Decision:**

Accept (poster)

**Comment:**

This paper proposes 4D-VLA, a vision–language–action pretraining framework that addresses two key sources of input inconsistency in embodied learning: coordinate system misalignment and temporal ambiguity. The method augments RGB with depth (or estimated geometry) and introduces a memory-bank sampling (MBS) strategy tailored to streaming VLA tasks. The authors also propose MV-Bench, a multi-view evaluation suite. Experiments on LIBERO, MV-Bench, and real robots show consistent improvements over baselines.
Strengths

The paper tackles a fundamental and under-explored challenge in VLA pretraining: the lack of consistent robot-centric coordinate frames in existing datasets. The integration of depth/geometry for coordinate alignment and the lightweight MBS strategy are well-motivated and effective for streaming policy inference. Experimental validation is broad, spanning simulation, multi-view benchmarks, and real-world robot rollouts. The rebuttal provides detailed clarifications, extensive additional comparisons (e.g., against LLARVA and ARM4R on RLBench), and transparent discussion of limitations.

Several reviewers questioned novelty, noting similarities to prior works such as 3D-VLA and ARM4R, and raised concerns about limited evaluation scope and task simplicity. Some reviewers felt the terminology (“coordinate system chaos,” “state chaos”) was overstated and could be made clearer. The method leverages calibrated camera intrinsics/extrinsics, which is an additional source of supervision compared to some baselines. Real-world experiments used only 50 demonstrations, which some reviewers felt was too small to fairly assess baselines.

The authors’ rebuttal was detailed and constructive. They clarified that: 1) Their use of depth is fundamentally different from 3D-VLA, emphasizing spatial consistency for action prediction rather than scene imagination. 2) Differences from ARM4R in pretraining paradigm, input modalities, and architectural focus. 3) Why MBS is lightweight and streaming-oriented compared to common video sampling methods. They also added further RLBench comparisons, demonstrating superior performance and efficiency, and acknowledged limitations (e.g., reliance on camera calibration, small-scale real-world evaluation).

Three reviewers (jMXX, GHr5, 7q7T) raised concerns but did not further engage after rebuttal. Reviewer rfTc engaged extensively, acknowledged the clarifications, and ultimately raised their score to borderline accept. While the paper received borderline or negative initial ratings, the ACs believe the rebuttal and subsequent clarifications address the main concerns to a satisfactory degree, including those raised by reviewers who did not respond during rebuttal. The core contribution of resolving coordinate frame inconsistency in VLA pretraining is promising, and the experimental evidence supports its effectiveness.

To conclude, this work makes a meaningful and timely contribution to the growing area of VLA pretraining and embodied foundation models. The ACs encourage the authors to incorporate their rebuttal clarifications, additional comparisons, and a more transparent discussion of limitations into the final version.